# Demystifying the Paradox of Importance Sampling with an Estimated History-Dependent Behavior Policy in Off-Policy Evaluation

**Hongyi Zhou** [1]  **Josiah P. Hanna** [2]  **Jin Zhu** [3]  **Ying Yang** [1]  **Chengchun Shi** [3]

## Abstract

This paper studies off-policy evaluation (OPE) in reinforcement learning with a focus on behavior policy estimation for importance sampling. Prior work has shown empirically that estimating a history-dependent behavior policy can lead to lower mean squared error (MSE) even when the true behavior policy is Markovian. However, the question of *why* the use of history should lower MSE remains open. In this paper, we theoretically demystify this paradox by deriving a bias-variance decomposition of the MSE of ordinary importance sampling (IS) estimators, demonstrating that history-dependent behavior policy estimation decreases their asymptotic variances while increasing their finite-sample biases. Additionally, as the estimated behavior policy conditions on a longer history, we show a consistent decrease in variance. We extend these findings to a range of other OPE estimators, including the sequential IS estimator, the doubly robust estimator and the marginalized IS estimator, with the behavior policy estimated either parametrically or non-parametrically.

## 1. Introduction

Off-policy evaluation (OPE) focuses on estimating the average return (sum of discounted rewards) of a specific decision policy, referred to as the target policy, by leveraging historical data collected under a potentially different policy, known as the behavior policy. OPE is vital in numerous domains where direct experimentation is impractical due to high costs, potential risks, or ethical concerns, such as in

healthcare (Murphy et al., 2001; Hirano et al., 2003), recommendation systems (Chapelle & Li, 2011) and robotics (Levine et al., 2020).

One widely used OPE method is importance sampling (IS, see e.g., Precup et al., 2000), which employs a reweighting approach to handle the distribution shift between the target policy and the behavior policy. This approach is straightforward: returns generated by the behavior policy are re-weighted based on the ratio of the probability of selecting actions under the target policy to that under the behavior policy. The re-weighted returns are then averaged to produce an unbiased estimator of the target policy's value. In the limit, as the number of trajectories increases, this estimator converges to the true value of the target policy. However, with finite samples, IS may exhibit high variance, causing considerable estimation error. Consequently, more advanced estimators have been proposed to lower its variance, including the doubly robust (DR) estimator (Jiang & Li, 2016; Thomas & Brunskill, 2016) and marginalized IS estimator (MIS, Liu et al., 2018). Despite its limitation, IS serves as a foundation for many OPE methods and is particularly valued in practice for its unbiasedness. It is also frequently used in off-policy learning algorithms, such as the proximal policy optimization algorithm (Schulman et al., 2017), which is widely used for fine-tuning large language models (Ouyang et al., 2022).

In practice, the behavior policy might be unknown and must be estimated from the historical data to construct the IS ratio. Paradoxically, IS with an estimated behavior policy results in an estimator with lower asymptotic variance and often lower finite-sample mean-squared error (MSE) compared to IS using the true behavior policy. This result has been shown in the statistics (Henmi et al., 2007), causal inference (Hirano et al., 2003; Rosenbaum & Rubin, 1983), multi-armed bandit (Xie et al., 2019a), and Markov decision process (MDP) policy evaluation (Hanna et al., 2021) literature. Furthering the paradox, Hanna et al. showed empirically that in MDPs where the true behavior policy is a first-order Markov-policy (action selection is conditioned only on the current state), the IS estimator's MSE could be lowered by estimating a higher-order Markov-policy where action selection is conditioned on a history of preceding

[1]Department of Mathematical Science, Tsinghua University, Beijing, China [2]Computer Sciences Department, University of Wisconsin – Madison, Madison, WI, USA [3]London School of Economics and Political Science, London, UK. Correspondence to: Chengchun Shi <c.shi7@lse.ac.uk>.

*Proceedings of the 42nd International Conference on Machine Learning*, Vancouver, Canada. PMLR 267, 2025. Copyright 2025 by the author(s).

*Table 1.* Impact of incorporating history-dependent IS ratios on bias and variance across various OPE estimators, where ↑ represents an increase, ↓ represents a decrease and → indicates no difference.

| METHOD | BIAS | VARIANCE |
|---|---|---|
| ORDINARY IS | ↑ | ↓ |
| SEQUENTIAL IS | ↑ | ↓ |
| DR (WITH A MISSPECIFIED $Q$) | ↑ | ↓ |
| DR (WITH A CORRECT $Q$) | ↑ | → |
| MARGINALIZED IS | ↑ | ↑ |

states (2021). However, the theoretical basis and generality of this finding was left as an open question.

In this work, we establish a comprehensive theoretical framework for analyzing OPE estimators with history-dependent IS ratios; refer to Table 1 for a quick summary of our findings. Our contributions are as follows:

- We demystify the aforementioned paradox for ordinary IS (OIS) estimators with history-dependent IS ratios by deriving a bias-variance decomposition of their MSEs. Our findings reveal that *in large samples, the variance component becomes the leading term in the MSE and can be reduced through history-dependent behavior policy estimation. Specifically, increasing the history-length, decreases the variance*.

- We also show that *there is no free lunch for using history-dependent IS ratios, as it comes at the price of increasing the bias of the resulting OPE estimator, which becomes non-negligible in finite samples.*

- We extend these findings to accommodate other variants of IS estimators, including the sequential IS (SIS), DR and MIS estimators, with the behavior policy estimated either parametrically, or non-parametrically. Interestingly, incorporating history-dependent IS ratios has different effects on the asymptotic variances of these estimators:

  (1) It *reduces* the asymptotic variance for SIS;
  (2) It leaves the asymptotic variance of DR *unchanged* when the Q-function is correctly specified, and *improves* the performance with a misspecified Q;
  (3) It *increases* the asymptotic variance for MIS.

## 2. Literature review on OPE

There is a huge literature on OPE in reinforcement learning (RL); see Uehara et al. (2022) for a recent review of existing methodologies. Current OPE methods can be grouped into four major categories:

- **Model-based methods**. These methods estimate an MDP model from the offline data and learn the policy value based on the estimated model (Gottesman et al., 2019; Yin & Wang, 2020; Wang et al., 2024).

- **Direct methods**. These methods estimate a value or Q-function to directly construct the policy value estimator (Sutton et al., 2008; Le et al., 2019; Feng et al., 2020; Luckett et al., 2020; Hao et al., 2021; Liao et al., 2021; Chen & Qi, 2022; Shi et al., 2022b; Li et al., 2023a; Liu et al., 2023; Bian et al., 2025).

- **IS methods**. This paper focuses on the family of IS estimators, which can be further classified into three types, according to the IS ratios used to reweight the rewards: (i) OIS, which employs the product of IS ratios from the initial time to the termination time to reweight the empirical return (Hanna et al., 2019; 2021); (ii) SIS, which also uses the product of IS ratios but applies a different product at each time to reweight the immediate reward (Thomas et al., 2015; Zhao et al., 2015; Guo et al., 2017); (iii) MIS, which uses an IS ratio on the marginal state-action distribution as a function of both the action and the state to adjust the reward (Liu et al., 2018; Nachum et al., 2019; Xie et al., 2019b; Dai et al., 2020; Wang et al., 2023; Zhou et al., 2023). In addition to these methods, several variants have been proposed to improve estimation accuracy, including incremental IS (Guo et al., 2017), conditional IS (Rowland et al., 2020), and state-based IS (Bossens & Thomas, 2024). These methods modify the IS ratio to enhance efficiency and are, in principle, similar to our proposal, which considers history-dependent behavior policy estimation as an alternative strategy for improving IS efficiency.

- **Doubly robust methods**. These methods combine the value or Q-function estimator used in direct methods and the IS ratios used in IS to construct the policy value estimator (Zhang et al., 2013; Jiang & Li, 2016; Thomas & Brunskill, 2016; Farajtabar et al., 2018; Bibaut et al., 2019; Tang et al., 2020; Uehara et al., 2020; Kallus & Uehara, 2020; 2022; Liao et al., 2022). A salient feature of these methods is their double-robustness property, which ensures the resulting policy value estimator's consistency as long as either one of the two nuisance function estimators to be correctly specified, not necessarily both. Several extensions of DR have been proposed in the literature, including triply robust estimators (Shi et al., 2021), semi-parametrically efficient estimators tailored to linear MDPs (Xie et al., 2023) and methods that estimate the difference in Q-functions (Cao & Zhou, 2024).

When the target policy itself is history-dependent, history-dependent behavior policy has been employed to correct the off-policy distributional shift (Kallus & Uehara, 2020).

However, in settings where the target policy is Markovian – a common scenario in MDPs due to the Markovian nature of the optimal policy (Puterman, 2014) – the effects of history-dependent behavior policy estimation on the accuracy of the resulting OPE estimator have been less explored. Hanna et al. (2019; 2021) demonstrated the possibility of lower MSE with a history-dependent behavior policy for evaluating Markov policies in MDPs. However, their work largely focused on estimating Markov behavior policies and left the justification for using history as an open question.

Our analysis significantly advances their analyses in the following ways: (i) We offer a bias-variance decomposition to theoretically demystify this paradox. (ii) We demonstrate that the variance varies monotonically with the number of preceding observations used to fit the behavior policy. (iii) As opposed to Hanna et al. (2019) and Hanna et al. (2021) whose focused on OIS estimator, our analysis extends to SIS, DR and MIS.

## 3. Building intuition: from bandits to MDPs

This section begins with a bandit example to introduce the OPE problem and IS estimators. This example serves to build intuition about how estimating a behavior policy that conditions on extra information than the true behavior policy can lead to a more accurate IS estimator. We next formulate the OPE problem in MDPs and describe the IS estimators for MDPs.

### 3.1. A bandit example

Consider a contextual bandit model $\mathcal{B} = (\mathcal{S}, \mathcal{A}, r)$ where $\mathcal{S}$ and $\mathcal{A}$ denote finite context and action spaces respectively, and $r : \mathcal{S} \times \mathcal{A} \to \mathbb{R}$ denotes a deterministic reward function. At each time, the agent observes certain contextual information $S \in \mathcal{S}$ and selects an action $A$ according to a behavior policy $\pi_b$ such that $\mathbb{P}(A = a|S) = \pi_b(a|S)$ for any $a \in \mathcal{A}$. Next, the environment responds by assigning a numerical reward $R$ to the agent, the conditional expectation of which, given the state-action pair, is equal to $r(S, A)$. Given $n$ independent and identically distributed (i.i.d.) copies of context-action-reward triplets, OPE aims to evaluate the expected reward the agent would have received under a certain target policy $\pi_e$, which may differ from $\pi_b$.

IS estimators are motivated by the change-of-measure theorem, which allows us to expresses the target policy's expected reward $v(\pi_e)$ based on the IS ratio and the observed reward as

$$v(\pi_e) = \mathbb{E}\Big[\frac{\pi_e(A|S)}{\pi_b(A|S)}R\Big]. \tag{1}$$

Assuming that both $\pi_b$ and $\pi_e$ are both context independent (i.e., $\pi_e(A|S) = \pi_e(A)$, $\pi_b(A|S) = \pi_b(A)$), we introduce three IS estimators that differ in their choice of the IS ratio:

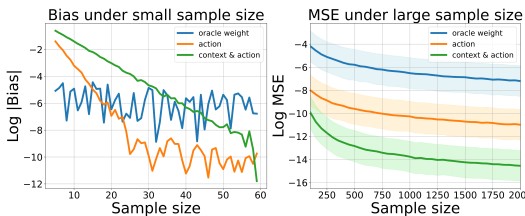

Figure 1. The left panel is log absolute bias of the three IS estimators. The right panel shows log MSE of three different estimators. Results are averaged over $10^4$ trials.

1. When $\pi_b$ is known to us, the first estimator uses the oracle IS ratio $\pi_e/\pi_b$ to estimate $v(\pi_e)$,

$$\widehat{v}^{\dagger}_{\text{IS}} = \mathbb{E}_n\Big[\frac{\pi_e(A)}{\pi_b(A)}R\Big],$$

where $\mathbb{E}_n$ denotes the empirical average over the $(S, A, R)$ triplets in the offline dataset. According to (1), it is immediate to see that $\widehat{v}^{\dagger}_{\text{IS}}$ is an unbiased estimator of $v(\pi_e)$[1].

2. Let $n(a)$ denote the number of occurrences of $A = a$ in the offline data. When $\pi_b$ remains unknown, it can be estimated by the sample mean estimator $\widehat{\pi}_b(a) = n(a)/n$, leading to the second IS estimator that employs a *context-agnostic* estimated IS ratio,

$$\widehat{v}^{\text{CA}}_{\text{IS}} = \mathbb{E}_n\Big[\frac{\pi_e(A)}{\widehat{\pi}_b(A)}R\Big].$$

3. Let $n(s, a)$ and $n(s)$ denote the number of occurrences of $(S = s, A = a)$ and $S = s$ in the offline data, respectively. When $\pi_b$ is unknown and not assumed to be context-independent, it is natural to estimate $\pi_b$ using $\widehat{\pi}_b(a|s) = n(s, a)/n(s)$, leading to a third estimator with a *context-dependent* estimated IS ratio

$$\widehat{v}^{\text{CD}}_{\text{IS}} = \mathbb{E}_n\Big[\frac{\pi_e(A)}{\widehat{\pi}_b(A|S)}R\Big].$$

Let $\text{MSE}_A(\bullet)$ denote the asymptotic MSE of a given estimator, obtained by removing errors that are high-order in the sample size $n$. The following lemma summarizes the performance of the three estimators in terms of their asymptotic MSEs.

**Lemma 1.** $MSE_A(\widehat{v}^{CD}_{IS}) \leq MSE_A(\widehat{v}^{CA}_{IS}) \leq MSE_A(\widehat{v}^{\dagger}_{IS})$. *The first equality hold if and only if the reward function $r$ is independent of the context $S$ whereas the second equality holds if and only if $\mathbb{E}(R|A) = 0$ almost surely.*

The two inequalities in Lemma 1 derive the following two seemingly paradoxical conclusions in the bandit setting:

---

[1]We will use the symbol † to denote estimators that use oracle IS ratios throughout the paper.

**Conclusion 1.** *Even when the behavior policy is known, using an estimated IS ratio can asymptotically improve the resulting IS estimator compared to the one using the oracle behavior policy.*

**Conclusion 2.** *Even when the true behavior policy is context-agnostic, incorporating context in estimating the IS ratio can asymptotically enhance the performance compared to using a context-agnostic ratio.*

Our numerical results, reported in Figure 1, empirically confirm these conclusions. As observed in the right panel, incorporating context-dependent estimated IS ratios substantially reduces the MSE. Given that the $y$-axis visualizes the log(MSE), even seemingly close log values can correspond to considerable differences in MSE values.

In what follows, we outline a sketch of the proof to demystify these results. The key insight is that *replacing the true behavior policy with its estimator in the IS ratio plays a similar role in adding an augmentation term to the IS estimator. This modification effectively transforms the resulting estimator into a DR estimator*, which is often more efficient than IS even in bandit settings (Tsiatis, 2006; Zhang et al., 2012; Dudík et al., 2014).

Specifically, it can be shown that $\widehat{v}_{\text{IS}}^{\text{CA}}$ and $\widehat{v}_{\text{IS}}^{\text{CD}}$ equal

$$\widehat{v}_{\text{IS}}^{\text{CA}} = \mathbb{E}_n \Big\{ \sum_a \pi_e(a)\widehat{r}(a) + \frac{\pi_e(A)}{\widehat{\pi}_b(A)}[R - \widehat{r}(A)] \Big\},$$

$$\widehat{v}_{\text{IS}}^{\text{CD}} = \mathbb{E}_n \Big\{ \sum_a \pi_e(a)\widehat{r}(S,a) + \frac{\pi_e(A)}{\widehat{\pi}_b(A|S)}[R - \widehat{r}(S,A)] \Big\},$$

respectively, where both $\widehat{r}(a)$ and $\widehat{r}(s,a)$ denote the sample mean estimators, obtained by averaging rewards across different contexts and/or actions.

In both expressions, the first terms within the curly brackets represent the direct method estimators for the policy value whereas the second terms serve as augmentation terms. The inclusion of these augmentation terms offers two advantages: (i) It debiases the bias inherent in the reward estimators, rendering the resulting OPE estimator asymptotically unbiased. (ii) It effectively reduces the variance of the OPE estimator by contrasting the observed reward with their predictor. Specifically, it can be shown that both expressions achieve no larger asymptotic variances than $\widehat{v}_{\text{IS}}^{\dagger}$ which uses the oracle IS ratio. Additionally, the variance reductions are likely substantial when the reward function differs significantly from 0. These discussions verify the assertions in Lemma 1.

In summary, our bandit example has revealed several intriguing conclusions that we aim to establish in MDPs. First, we will demonstrate that Conclusion 1 remains valid across a range of IS-type estimators with history-dependent behavior policy estimators in MDPs. Second, we will expand on Conclusion 2 by demonstrating that estimating a behavior policy that conditions on history leads to more accurate OPE estimators in large samples – even when the true behavior policy does not condition on more than the immediate preceding state. Finally, the above theoretical analysis did not consider the biases of IS estimators. As depicted in the left panel of Figure 1, incorporating history-dependent behavior policy estimation can increase bias in small samples. In our forthcoming analysis of MDPs, we will carefully examine the finite-sample biases of different IS estimators.

### 3.2. OPE in MDPs

**Markov decision processes**. This paper focuses on a finite-horizon MDP model $\mathcal{M}$ characterized by a state space $\mathcal{S}$, an action space $\mathcal{A}$, a transition kernel $\mathcal{P}: \mathcal{S} \times \mathcal{S} \times \mathcal{A} \to \mathbb{R}$, a reward function $r: \mathcal{S} \times \mathcal{A} \to \mathbb{R}$ and a finite horizon $T < \infty$. Consider a trajectory $H := (S_0, A_0, R_0, \ldots, S_T, A_T, R_T)$ generated in $\mathcal{M}$. These data are generated as follows:

- At each time, suppose the environment arrives at a given state $S_t \in \mathcal{S}$;

- The agent then selects an action $A_t \in \mathcal{A}$ according to a behavior policy $\pi_b(\bullet|S_t)$;

- Next, the environment provides an immediate reward to the agent whose expected value is specified by the reward function $r(S_t, A_t)$;

- Finally, the environment transits into a new state $S_{t+1}$ at time $t+1$ according to the transition function $\mathcal{P}(\bullet|S_t, A_t)$.

This process repeats until the termination time, $T$, is reached.

**Common IS-type estimators**. Given an offline dataset with $n$ i.i.d. trajectories, the objective of OPE is to learn the expected cumulative reward $v(\pi_e) = \mathbb{E}_{\pi_e}(\sum_{t=0}^T \gamma^t R_t)$ under a different target policy $\pi_e$, where $\gamma \in (0, 1]$ denotes the discount factor and $\mathbb{E}_{\pi_e}$ denotes the expectation assuming the actions are assigned according to $\pi_e$.

Let $\mathbb{E}_n$ denote the empirical average operator over the $n$ trajectories in the offline dataset and $\lambda_t$ denote the product of IS ratios $\prod_{k=1}^t \frac{\pi_e(A_k|S_k)}{\pi_b(A_k|S_k)}$ up to time $t$. Below, we detail the definitions of the three types of IS estimators introduced in Section 2, along with the DR estimator which also employs IS ratios for OPE:

1. **OIS** serves as the most foundational estimator. It applies a single weight $\lambda_T$ to reweight the entire empirical return $G_T = \sum_{t=0}^T \gamma^t R_t$, leading to $\widehat{v}_{\text{OIS}}^{\dagger} = \mathbb{E}_n(\lambda_T G_T)$.

2. **SIS** modifies OIS by applying a time-dependent ratio $\lambda_t$ to reweight each reward $R_t$, resulting in $\widehat{v}_{\text{SIS}}^{\dagger} = \mathbb{E}_n(\sum_{t=0}^T \gamma^t \lambda_t R_t)$. This adjustment reduces the variance associated with the product of IS ratios since, at each time $t$, only ratios up to that time are used.

3. **DR** further employs an estimated Q-function to reduce the variance of SIS. Specifically, let $Q_t^{\pi_e}(s, a)$ denote the Q-function under the target policy, which measures the cumulative reward starting from a given state-action pair

$$Q_t^{\pi_e}(s, a) = \sum_{k=t}^{T} \gamma^{k-t} \mathbb{E}_{\pi_e}(R_k | A_t = a, S_t = s).$$

Given a Q-function estimator $Q = \{Q_t\}_t$ for $\{Q_t^{\pi_e}\}_t$, DR is defined by

$$\widehat{v}_{\text{DR}}^{\dagger} = \mathbb{E}_n \Big\{ \sum_{t=0}^{T} \Big[ \lambda_t \gamma^t \Big( R_t - Q_t(S_t, A_t) \Big) + \lambda_{t-1} \gamma^t \sum_a Q_t(S_t, a) \pi_e(a|S_t) \Big] \Big\},$$

with the convention that $\lambda_{-1} = 1$. Since $\widehat{v}_{\text{DR}}^{\dagger}$ employs the oracle IS ratio and leverages the double-robustness property, it remains consistent regardless of whether the Q-function is correctly specified.

4. **MIS** further reduces the variances of the aforementioned three estimators by replacing $\lambda_t$ – which is known to suffer from the curse of horizon (Liu et al., 2018) – with an MIS ratio given by $w_t = d_{\pi_e,t}(S_t, A_t)/d_{\pi_b,t}(S_t, A_t)$ where $d_{\pi_e,t}(\cdot)$ and $d_{\pi_b,t}(\cdot)$ are the marginal distributions of $(S_t, A_t)$ induced by policies $\pi_e$ and $\pi_b$, respectively. This leads to $\widehat{v}_{\text{MIS}}^{\dagger} = \mathbb{E}_n(\sum_{t=0}^{T} \gamma^t w_t R_t)$.

We will investigate the theoretical properties of these estimators in the next two sections.

## 4. Demystifying the paradox in MDPs

In this section, we conduct a rigorous theoretical analysis to evaluate the impact of replacing the oracle behavior policy with an estimated history-dependent behavior policy for OPE. Our analysis accommodates all four estimators discussed in Section 3.2.

Although $\pi_b$ is a Markov policy, historical observations can still be utilized to estimate it. In particular, we define the following estimator that uses $k$-step state-action history $H_{t-k:t} = (S_{t-k}, A_{t-k}, \ldots, S_{t-1}, A_{t-1}, S_t)$,

$$\widehat{\pi}_b^{(k)} = \arg\max_{\pi \in \Pi_k} \mathbb{E}_n \Big[ \sum_{t=0}^{T} \log \pi(A_t | H_{t-k:t}) \Big],$$

for some policy class $\Pi_k$ that satisfies the following monotonicity assumption:

**Assumption 1** (Monotonicity). $\Pi_0 \subseteq \Pi_1 \subseteq \Pi_2 \subseteq \cdots$.

Most commonly used policy classes based on logistic regression models or neural networks satisfy Assumption 1. We discuss this assumption in greater detail in Appendix C.2 and impose the following assumptions.

**Assumption 2** (Realizability). There exists some $\theta^* \in \Pi_0$ such that $\pi_b = \pi_{\theta}^*$.

**Assumption 3** (Bounded rewards). There exists some constant $R_{\max} < \infty$ such that $|R_t| \leq R_{\max}$ almost surely for any $t$.

**Assumption 4** (Coverage). There exist some constants $\varepsilon > 0, C \geq 1$ such that all policy functions $\pi_\theta$ are lower bounded by $\varepsilon$, and $\pi_e(s, a)/\pi_\theta(s, a) \leq C$ holds for all state-action pair $(s, a)$.

**Assumption 5** (Differentiability). All policies $\pi_\theta$ are twice differentiable with respect to the parameter $\theta$, and both its first and second derivatives are uniformly bounded.

**Assumption 6** (Non-singularity). The Fisher information matrix of $\theta^*$, denoted by $I(\theta^*)$, is non-singular.

We make a few remarks. First, realizability assumes that the policy class $\Pi_0$ is rich enough to cover $\pi_b$. It is a common assumption in machine learning (Shalev-Shwartz & Ben-David, 2014). It will be relaxed in Section 5 by permitting a nonzero approximation error. Second, the bounded rewards and coverage conditions are frequently assumed in the RL and OPE literature (see e.g., Chen & Jiang, 2019; Fan et al., 2020; Kallus & Uehara, 2022). Finally, Assumptions 5 and 6 are widely imposed in statistics to establish the theoretical properties of maximum likelihood estimators (see e.g., Casella & Berger, 2024).

### 4.1. Ordinary IS estimator

Recall from Section 3.2 that $\widehat{v}_{\text{OIS}}^{\dagger}$ denotes the OIS estimator with the oracle IS ratio $\lambda_T$. Let $\widehat{v}_{\text{OIS}}(k)$ denote the version that uses the $k$-step state-action history to compute the behavior policy estimator $\widehat{\pi}_b^{(k)}$ and plugs it into $\lambda_T$ to construct the ratio estimator $\widehat{\lambda}_T(k)$,

$$\widehat{v}_{\text{OIS}}(k) = \mathbb{E}_n[\widehat{\lambda}_T(k) G_T].$$

The following theorem establishes the theoretical properties of these estimators.

**Theorem 2.** *Assume Assumptions $1 - 6$ hold. Then*

$$\text{MSE}(\widehat{v}_{\text{OIS}}(k)) = \frac{1}{n} \text{Var}\Big( \text{Proj}_{\mathbb{T}(k)}(\lambda_T G_T) \Big) + O\Big( \frac{(k+1)C^{2T} R_{max}^2}{n^{3/2} \varepsilon^2} \Big),$$

(2)

*where $\mathbb{T}(k)$ denotes the space of mean zero random variables that is orthogonal to the tangent space spanned by the score vector*

$$s(H, k; \theta^*) = \frac{\partial}{\partial \theta} \sum_{t=0}^{T} \log \pi_\theta(A_t | H_{t-k:t}) \Big|_{\theta=\theta^*},$$

*and $\text{Proj}_{\mathbb{T}(k)}(\bullet)$ denotes the projection of a given random variable onto the space of $\mathbb{T}(k)$; refer to Appendix C.2 for*

*the detailed definitions. Moreover, for any $k' < k$, we have*

$$
\begin{aligned}
\mathrm{Var}\Big(\mathrm{Proj}_{\mathbb{T}(k)}(\lambda_T G_T)\Big) &= \mathrm{Var}\Big(\mathrm{Proj}_{\mathbb{T}(k')}(\lambda_T G_T)\Big) \\
&-\mathrm{Var}\Big(\mathrm{Proj}_{\mathbb{T}(k')}(\lambda_T G_T) - \mathrm{Proj}_{\mathbb{T}(k)}(\lambda_T G_T)\Big).
\end{aligned}
\tag{3}
$$

Theorem 2 has a number of important implications:

1. Equation (2) obtains a bias-variance decomposition for the MSE of $\widehat{v}_{\mathrm{OIS}}(k)$. In particular, the first term on the right-hand-side (RHS) of (2) corresponds to its asymptotic variance, which is of the order $O(n^{-1})$, whereas the second term upper bounds its finite-sample bias, which decays to zero at a faster rate as $n$ increases. Additionally, it is well known that the variances of IS-type estimators grow exponentially fast with the time horizon (see, e.g., Liu et al., 2018). Our error bound reveals that when using estimated IS ratios, the same curse of horizon applies to the bias, which includes a factor of $C^{2T}$ for some $C \geq 1$, where $C = 1$ if and only if the behavior policy matches the target policy, meaning there is no off-policy distributional shift at all.

2. In large samples, the asymptotic variance term becomes the dominating factor. This term equals the variance of $\mathbb{E}_n[\mathrm{Proj}_{\mathbb{T}(k)}(\lambda_T G_T)]$. *Thus, incorporating history-dependent behavior policy estimation into OIS estimators can be interpreted as a projection that projects the empirical return into a more constrained space for variance reduction.* This interpretation aligns with our perspective on transforming IS estimators with estimated ratios into DR estimators, as illustrated in the bandit example (see Section 3.1), since DR can be viewed as projecting an IS estimator onto a specific augmentation space to improve efficiency (Tsiatis, 2006). Notice that the projected variable $\mathrm{Proj}_{\mathbb{T}(k)}(\lambda_T G_T)$ achieves a smaller variance than $\lambda_T G_T$ itself, our result thus covers Corollary 2 in Hanna et al. (2021), suggesting that replacing the true behavior policy with its estimate reduces the asymptotic variance of the resulting OIS estimator.

3. Additionally, according to (3), the variance term is a monotonically non-decreasing function with respect to the history-length, which in turn demonstrates the advantage of estimating a high-order Markov policy over a first-order policy in large samples. Mathematically, this can again be interpreted through projection: *the longer the history-length, the more restrictive the constrained space used to project the empirical return, leading to greater asymptotic efficiency.*

4. In small samples, particularly in settings with long horizons, the bias term becomes non-negligible and increases exponentially with the horizon. To the contrary, the oracle estimator $\widehat{v}_{\mathrm{OIS}}^{\dagger}$ is unbiased. This illustrates the risk of employing history-dependent behavior policy estimation in small samples.

Based on the aforementioned discussion, the following corollary is immediate from Theorem 2.

**Corollary 3.** *Let $k$ and $k'$ be two positive integers satisfying $k' \leq k$. Under Assumptions 1 – 6, we have*

$$
\mathrm{MSE}_A(\widehat{v}_{\mathrm{OIS}}(k)) \leq \mathrm{MSE}_A(\widehat{v}_{\mathrm{OIS}}(k'))
$$

To summarize, Theorem 2 formally establishes the bias-variance trade-off in history-dependent behavior policy estimation: it decreases the asymptotic variance of the OIS estimator at the cost of increasing the finite-sample bias. Furthermore, a longer history length results in a greater reduction in variance.

### 4.2. Sequential IS estimator

Let $\widehat{\lambda}_t(k)$ denote the estimator for $\lambda_t$ by replacing the oracle behavior policy with its estimator $\widehat{\pi}_b^{(k)}$. We define $\widehat{v}_{\mathrm{SIS}}(k)$ as a variant of the oracle SIS estimator $\widehat{v}_{\mathrm{SIS}}^{\dagger}$ constructed based on $\{\widehat{\lambda}_t(k)\}_t$. The following theorem obtains a similar bias-variance decomposition for its MSE.

**Theorem 4.** *Assume Assumptions 1 – 6 hold. Then*

$$
\begin{aligned}
\mathrm{MSE}(\widehat{v}_{\mathrm{SIS}}(k)) &= \frac{1}{n}\mathrm{Var}\Big(\mathrm{Proj}_{\mathbb{T}(k)}(\sum_{t=0}^{T}\lambda_t \gamma^t R_t)\Big) \\
&+ O\Big(\frac{(k+1)C^{2T}R_{max}^2}{n^{3/2}\varepsilon^2}\Big).
\end{aligned}
\tag{4}
$$

*In addition, the first term on the RHS of (2) is non-decreasing with respect to $k$.*

Recall that the oracle SIS estimator $\widehat{v}_{\mathrm{SIS}}^{\dagger}$ is given by $\mathbb{E}_n(\sum_{t=0}^{T}\lambda_t \gamma^t R_t)$. Similar to OIS, Theorem 4 suggests that using an estimated behavior policy will lower the MSE of the resulting SIS estimator in large samples through projection. Meanwhile, the longer the history-length, the lower the asymptotic MSE, leading to the following corollary.

**Corollary 5.** *Let $k$ and $k'$ be two positive integers satisfying $k' \leq k$. Then under Assumptions 1 – 6,*

$$
\mathrm{MSE}_A(\widehat{v}_{\mathrm{SIS}}(k)) \leq \mathrm{MSE}_A(\widehat{v}_{\mathrm{SIS}}(k'))
$$

However, estimating the behavior policy can introduce significant biases in small samples and long horizons, the magnitudes of which are given by the second term in (4).

### 4.3. Doubly robust estimator

Consider the following DR estimator constructed based on the history-dependent IS ratio $\widehat{\lambda}_t(k)$,

$$
\begin{aligned}
\widehat{v}_{\mathrm{DR}}(k) &= \mathbb{E}_n\Big\{ \sum_{t=0}^{T} \lambda_t \gamma^t \left( R_t - Q_t(S_t, A_t) \right) \\
&\quad + \lambda_{t-1} \gamma^t \sum_a Q_t(S_t, a) \pi_e(a|S_t) \Big\},
\end{aligned}
$$

with a pre-specified Q-function which is required to satisfy the following assumption:

**Assumption 7** (Boundedness). There exists some $U_{\max} < \infty$ such that the absolute value of $U_t = R_t - Q_t(S_t, A_t) + \gamma Q_{t+1}(a, S_{t+1})$ is upper bounded by $U_\infty$ almost surely for any $t$.

Assumption 7 corresponds to a version of the boundedness condition in Assumption 3 tailored for DR estimators. The constant $U_{\max}$ is expected to be much smaller than $R_{\max}$ with a well-chosen Q-function. In particular, when the Q-function is correctly specified, $U_t$ corresponds to the absolute value of the Bellman residual, which tends to concentrate more closely around zero than $R_t$.

**Theorem 6.** *Assume Assumptions 1, 2, 5 – 7 hold. Then,*

$$
\begin{aligned}
\mathrm{MSE}(\widehat{v}_{\mathrm{DR}}(k)) &= \frac{1}{n} \mathrm{Var}\Big( \mathrm{Proj}_{\mathbb{T}(k)}\big(\sum_{t=0}^{T} \lambda_t \gamma^t U_t\big) \Big) \\
&\quad + O\Big( \frac{(k+1)C^{2T} U_{max}^2}{n^{3/2}\varepsilon^2} \Big).
\end{aligned} \tag{5}
$$

*In addition, the first term on the RHS of (5) is non-decreasing with respect to $k$. However, when the Q-function is correctly-specified, this term becomes a constant function of $k$.*

We make two remarks regarding Theorem 6:

1. The bias-variance decomposition in (5) closely resembles that of SIS, with the key difference being that the reward $R_t$ and its bound $R_{\max}$ in (4) are replaced with $U_t$ and $U_{\max}$, respectively. With a well-specified Q-function, $U_t$ is expected to exhibit lower variability than $R_t$, and $U_{\max}$ can be significantly smaller than $R_{\max}$. This highlights the advantages of history-dependent DR estimators over SIS: they not only improve asymptotic variance but also reduce finite-sample bias.

2. However, the second part of Theorem 6 indicates that, unlike OIS or SIS, history-dependent behavior policy estimation may not further reduce asymptotic variance when the Q-function is correctly specified. This is intuitive, as in such cases, the DR estimator is known to achieve certain efficiency bounds (Jiang & Li, 2016;

Kallus & Uehara, 2020). If the estimator is already efficient, history-dependent behavior policy estimation cannot provide additional gains. On the other hand, when the Q-function is misspecified, there remains room for improvement, and history-dependent estimators can improve the estimation accuracy.

The following corollary is again an immediate application of Theorem 5.

**Corollary 7.** *Under Assumptions 1, 2, 5 – 7, we have for any $k' \le k$ that*

$$
\mathrm{MSE}_A(\widehat{v}_{\mathrm{DR}}(k)) \le \mathrm{MSE}_A(\widehat{v}_{\mathrm{DR}}(k')).
$$

*The equation holds when the Q-function is correctly specified. In that case, we have $\mathrm{MSE}_A(\widehat{v}_{\mathrm{DR}}(k)) = \mathrm{MSE}_A(\widehat{v}_{\mathrm{DR}}^\dagger)$ for any $k$.*

### 4.4. Marginalized importance sampling estimator

A key step in constructing the MIS estimator lies in the estimation of the MIS ratio. Unlike the previously discussed ratios $\{\lambda_t\}_t$, which can be known in settings such as randomized studies, the MIS ratio depends on the marginal state distribution and is typically unknown, even when the behavior policy is given.

In the literature, several methods have been developed to estimate the MIS ratio, such as minimax learning (Uehara et al., 2020) and reproducing kernel Hilbert space (RKHS)-based methods (Liao et al., 2022). To simplify the analysis, we focus on using linear function approximation in this paper, which parameterizes each $w_t$ by $\phi_t^\top(S_t, A_t)\alpha_t$, for some state-action features $\phi_t$. Adapting Example 2 from Uehara et al. (2020) to the finite-horizon setting, we derive the following closed-form expression for the estimator $\widehat{\alpha}_0$,

$$
\widehat{\alpha}_0 = \widehat{\Sigma}_0^{-1} \mathbb{E}_n\Big[ \sum_a \pi_e(a|S_0)\phi_0(S_0, a)) \Big],
$$

where $\widehat{\Sigma}_t = \mathbb{E}_n\Big[ \phi_t(S_t, A_t)\phi_0^\top(S_0, A_0) \Big]$, and the following recursive formulas for computing $\widehat{\alpha}_t$,

$$
\widehat{\alpha}_t = \widehat{\Sigma}_t^{-1} \mathbb{E}_n\Big[ \sum_a \pi_e(a|S_t)\phi_t(S_t, a))\phi_{t-1}^\top(S_{t-1}, A_{t-1}) \Big]\widehat{\alpha}_{t-1}.
$$

The estimated MIS ratios $\{\widehat{w}_t = \phi_t^\top(S_t, A_t)\widehat{\alpha}_t\}_t$ are then plugged into the oracle estimator $\widehat{v}_{\mathrm{MIS}}^\dagger$ to compute $\widehat{v}_{\mathrm{MIS}}(0)$.

Alternatively, the $k$-step history $H_{t-k:t}$ can be used to construct a history-dependent MIS ratio $w_t(k) = \mathbb{E}(\lambda_t | H_{t-k:t}, A_t)$. This ratio can be interpreted as a conditional IS ratio (Rowland et al., 2020) with $H_{t-k:t}$ and $A_t$ being the conditioning variable. It is also closely related to the incremental IS (INCRIS) ratio proposed by Guo et al. (2017), but differs by incorporating an additional MIS ratio for $S_{t-k}$.

For estimation, $w_t(k)$ can be parameterized similarly to $w_t$, using $k$-step features $\phi_t(k)$ as a function of $H_{t-k:t}$ and $A_t$, with parameters estimated in a manner similar to those for $w_t$. However, unlike IS and DR, incorporating a history-dependent MIS ratio may increase the MSE of the resulting MIS estimator, denoted by $\widehat{v}_{\text{MIS}}(k)$. Additionally, the longer the history-length, the worsen the performance. We summarize these results in the following theorem.

**Theorem 8.** *Let* $\widehat{v}_{\text{MIS}}(k)$ *be the* MIS *estimator with $k$-step history: Then, under regularity conditions specified in Appendix C.2, for any $k' < k$,*

$$\text{MSE}_A(\widehat{v}_{\text{MIS}}(k')) \leq \text{MSE}_A(\widehat{v}_{\text{MIS}}(k)).$$

To appreciate why Theorem 8 holds, notice that by setting $k$ to the horizon $T$, $w_t(k)$ is reduced to the $\lambda_t$, and the resulting estimator is reduced to SIS, which suffers from the curse of horizon and is known to be less efficient than MIS. More generally, similar to , increasing the history-length leads to a more variable IS ratio, thus increasing the MSE.

## 5. Extensions to cases where the behavior policy is estimated nonparametrically

Our analysis so far focuses on using parametric models to estimate the behavior policy or IS ratio. In practical applications, nonparametric estimation of the behavior policy can be desirable to avoid the potential misspecification of the parametric model. This motivates us to investigate the performance of history-dependent OPE estimators with nonparametrically estimated behavior policy.

A common nonparametric approach is to approximate the policy set $\Pi$ using a sequence of sieve spaces $\Pi_n$. Below, we demonstrate that, under certain regularity conditions (detailed in Appendix C.3), similar to the parametric case, replacing the true behavior policy with an estimated behavior policy within the sieve space lowers the asymptotic variance of the resulting OPE estimator.

Specifically, we assume the policy class $\Pi$ can be represented by $\{\pi(H_{t-k:t}; \theta), \theta \in \Theta\}$ with an infinite-dimensional Hilbert space $\Theta$. Let $\Theta_1 \subseteq \ldots \Theta_n \subseteq \Theta_{n+1} \ldots \subseteq \Theta$ be a sequence of finite-dimensional sieve spaces. For a given sample size $n$, we compute the estimator $\widehat{\theta}_n$ by maximizing the log-likelihood function in the sieve space $\Theta_n$,

$$\widehat{\theta}_n(k) = \arg\max_{\theta \in \Theta_n} \mathbb{E}_n \Big[ \sum_{t=0}^{T} \log \pi_\theta(A_t | H_{t-k:t}) \Big].$$

Let $\widehat{v}_{\text{OIS}}(k)$, $\widehat{v}_{\text{SIS}}(k)$ and $\widehat{v}_{\text{DR}}(k)$ denote the OIS, SIS and DR estimators, respectively, each constructed based on the estimated behavior policy $\pi(H_{t-k:t}; \widehat{\theta}_n(k))$. We summarize our results as follows.

**Theorem 9.** *Under Assumptions 8 - 13 defined in Appendix C.3, we have*

$$\begin{aligned}
\text{MSE}_A(\widehat{v}_{\text{OIS}}(k)) &\leq \text{MSE}_A(\widehat{v}_{\text{OIS}}^\dagger), \\
\text{MSE}_A(\widehat{v}_{\text{SIS}}(k)) &\leq \text{MSE}_A(\widehat{v}_{\text{SIS}}^\dagger), \\
\text{MSE}_A(\widehat{v}_{\text{DR}}(k)) &\leq \text{MSE}_A(\widehat{v}_{\text{DR}}^\dagger).
\end{aligned}$$

Theorem 9 demonstrates the advantages of OPE estimators with nonparametrically estimated behavior policies in large samples. While similar results have been established in the literature (see e.g., Hanna et al., 2021), they primarily focused on the OIS estimator using parametric estimation of the behavior policy and required the realizability assumption (see Assumption 2). In contrast, Theorem 9 relaxes the realizability by allowing the approximation error to decay to zero at a rate of $o(n^{-1/4})$ (see Assumption 9), which is much slower than the parametric $n^{-1/2}$-rate. Nonetheless, we demonstrate that the resulting OPE estimators still converge at the parametric rate, which is central to establish their MSEs. This faster convergence rate occurs because the policy value is a smooth functional of the sieve estimator, and "smoothing" inherently improves the convergence rate. While similar findings have been documented in classical statistics literature for nonparametric regression problems (Shen, 1997; Newey et al., 1998), these phenomena have not been less explored in OPE and RL. One exception is Shi et al. (2023), who considered the direct method estimator but did not study history-dependent behavior policy estimation.

## 6. Numerical studies

Our experiment compares several history-dependent IS estimators in the CartPole environment (Brockman et al., 2016). Specifically, we consider the following three estimators: SIS, DR with a misspecified Q-function, and MIS.

As shown in Figure 2, all three estimators' MSEs decrease with the sample size, suggesting their consistencies. For SIS and DR with misspecified Q-functions, replacing the oracle behavior policy with a history-dependent estimator generally reduces their MSEs in large samples. Additionally, performance improves with longer history-length. However, for MIS estimators, the performance consistently worsens as we increase the history-length to estimate the MIS ratio. Finally, it is also apparent that history-dependent estimators generally suffer from larger biases compared to those using an oracle behavior policy. These empirical results verify our theoretical findings.

In Appendix B, we further expand our numerical experiments to more complex MuJoCo environments, including (i) Inverted Pendulum, featuring a continuous action space; (ii) Double Inverted Pendulum, characterized by a higher-dimensional state space; (iii) Swimmer, an environment

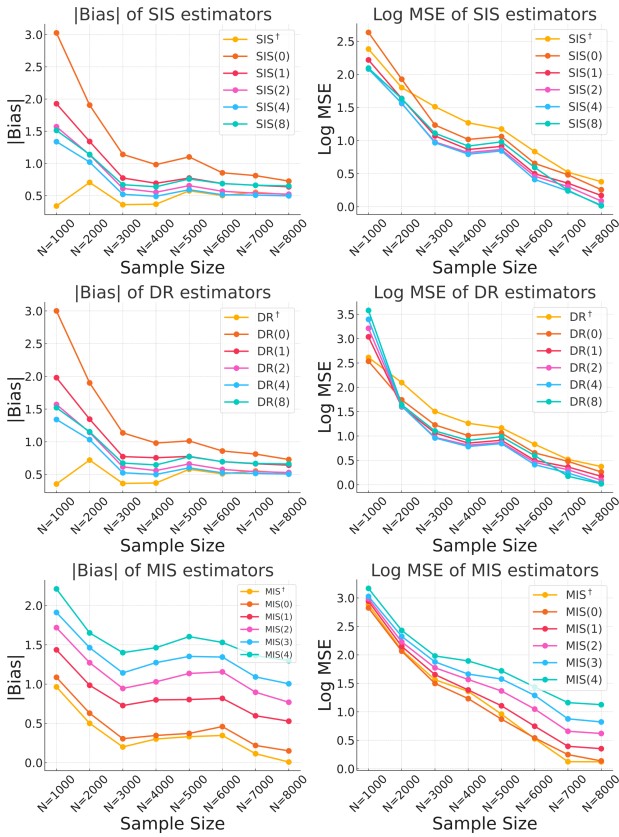

*Figure 2.* Absolute bias (left panel) and log MSE (right panel) of three OPE estimators: SIS (top panel), DR (middle panel), MIS (top panel). The results are averaged over 50 simulations.

with substantially different dynamics compared to the other two. The detailed results are deferred to Appendix B.

## 7. Discussion

This paper demystifies the paradox concerning the impact of history-dependent behavior policy estimation on IS-type OPE estimators by establishing a bias-variance decomposition of their MSEs. Our analysis reveals a trade-off in the choice of history-length for estimating the behavior policy: increasing the history-length reduces the estimator's asymptotic variance, but can increase its finite-sample bias. Therefore, selection of history length is crucial for applying our theory to practice.

In this section, we propose some practical guidance on the selection of history length when estimating behavior policy. Specifically, motivated by the bias-variance trade-off, we propose to select the history length that minimizes

$$h^* = \arg\min_h [2n\widehat{\mathrm{Var}}(h) - h\log(n)],$$

where $\widehat{\mathrm{Var}}(h)$ denotes variance estimator computed via the sampling variance formula or bootstrap, $k\log(n)$ is the Bayesian information criterion (BIC, Schwarz, 1978) penalty preventing selecting long history without substantial reduction of the variance. Our simulation studies (not reported in the paper) demonstrate strong empirical performance of this history selection method.

To conclude this paper, we note that the OPE literature has been growing rapidly in recent years, expanding into several directions, including the investigation of partially observable environments (Uehara et al., 2023; Hu & Wager, 2023), heavy-tailed rewards (Xu et al., 2022; Liu et al., 2023; Rowland et al., 2023; Zhu et al., 2024; Behnamnia et al., 2025) and unmeasured confounders (Kallus & Zhou, 2020; Namkoong et al., 2020; Tennenholtz et al., 2020; Nair & Jiang, 2021; Shi et al., 2022a; Wang et al., 2022; Bruns-Smith & Zhou, 2023; Xu et al., 2023; Bennett & Kallus, 2024; Shi et al., 2024; Yu et al., 2024). Our proposal is related to a growing line of research that investigates optimal experimental design for OPE (Hanna et al., 2017; Mukherjee et al., 2022; Wan et al., 2022; Li et al., 2023b; Liu & Zhang, 2024; Liu et al., 2024; Sun et al., 2024; Wen et al., 2025). These works focus on designing optimal behavior policies prior to data collection to improve OPE accuracy whereas our proposal considers estimating behavior policies after data collection for the same purpose. The work of Liu & Zhang (2024) is particularly related as the behavior policy is computed from offline data before being run to collect more data. Both approaches share the most fundamental goal of enhancing OPE by learning behavior policies - whether for data collection or retrospective estimation.

## Acknowledgement

Hongyi Zhou's and Ying Yang's research was partially supported by NSFC 12271286 & 11931001. Hongyi Zhou's research was also partially supported by the China Scholarship Council. Chengchun Shi's and Jin Zhu's research was partially supported by the EPSRC grant EP/W014971/1. Josiah Hanna acknowledges support from NSF (IIS-2410981), American Family Insurance through a research partnership with the University of Wisconsin—Madison's Data Science Institute, the Wisconsin Alumni Research Foundation, and Sandia National Labs through a University Partnership Award. The authors thank the anonymous referees and the area chair for their insightful and constructive comments, which have led to a significantly improved version of the paper.

## Impact statement

This paper provides a theoretical foundation for using history-dependent behavior policy estimators for OPE in re-

inforcement learning. Our research reveals that while these estimators may decrease accuracy with small sample sizes, they significantly improve estimation accuracy as sample size increases. This insight clarifies when and how historical data should be integrated into behavior policy estimation, enhancing the effectiveness and reliability of various off-policy estimators across different applications. Our work primarily engages in theoretical analysis and does not directly interact with or manipulate real-world systems. Consequently, it is unlikely to have negative societal consequences.

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

# A. Details of experiments

**Bandit example in Section 3.1.** In our illustrative example, we set the context space $\mathcal{S} = \{0, 1\}$, the action space $\mathcal{A} = \{0, 1\}$. The target policy $\pi_b$ is set as

$$\pi_e(1) = \mathbb{P}_e(A = 1) = 0.4, \qquad \pi_e(0) = \mathbb{P}_e(A = 0) = 0.6.$$

The behavior policy is set as

$$\pi_b(1) = \mathbb{P}_b(A = 1) = 0.3, \qquad \pi_b(0) = \mathbb{P}_b(A = 0) = 0.7.$$

Both the target and behavior policies are independent of context information. The context information $S$ follows a Bernoulli distribution with parameter $0.5$, that is,

$$\mathbb{P}(S = 0) = \mathbb{P}(S = 1) = 0.5.$$

Given context information $S$ and action $A$, the reward is a random variable with mean $10a + 0.1(1 + 2s)$. Therefore, the reward function is a deterministic function defined as

$$r(s, a) = 10a + 0.1(1 + 2s).$$

For the illustrative example, we can derive the closed-form expression of the policy's value, which is 4.2.

**Numerical experiments in Section 6.** In Cartpole environment, the state space $\mathcal{S}$ is a subset of $\mathbb{R}^4$. For any $s \in \mathcal{S}$, $s$ is characterized by four elements $(x, \dot{x}, \theta, \dot{\theta})$, where $x, \dot{x}$ are the position and velocity of the cart, $\theta, \dot{\theta}$ are the angle and angle velocity of the pole with the vertical axis. The behavior policy and the target policy are set as

$$\pi_b(a|s) \sim \text{Bernoulli}(p_b), \text{ where } p_b = 1/\left(1 + \exp(10\theta)\right);$$
$$\pi_e(a|s) \sim \text{Bernoulli}(p_e), \text{ where } p_e = 1/\left(1 + \exp(20\theta)\right).$$

Given $s = (x, \dot{x}, \theta, \dot{\theta})$, the reward is defined as $R = (2 - x/x_{\max})(2 - \theta/\theta_{\max}) - 1$. The maximum episode length is set as 200. We use a logistic regression model to estimate the behavior policy. The state transition model is set as the physical system implemented in CartPole environment in the `gym` library. And the initial state are uniformly drawn from $[-0.05, 0.05]^4$.

We use a Monte Carlo (MC) procedure to approximate the true value of target policy. Specifically, we run the deploy the target policy to the simulator and get a empirical cumulative reward $\widehat{v}_{\text{MC}}^{(l)}$. The procedure is repeated $L$ times, and the MC estimator is given by

$$\widehat{v}_{\text{MC}} = \frac{1}{L} \sum_{l=1}^{L} \widehat{v}_{\text{MC}}^{(l)}.$$

In our experiments, we set $L = 10^6$ and the value of $\widehat{v}_{\text{MC}}$ is 92.91.

# B. Additional experiment results

In this section, we examine the impact of using history-dependent behavior policies in the OIS estimator across three MuJoCo environments: (i) **Inverted Pendulum**; (ii) **Double Inverted Pendulum** and (iii) **Swimmer**.

For both Inverted Pendulum and Double Inverted Pendulum, the behavior policy is modeled using a transformed Beta distribution. Specifically, we set the action to $2Z - 1$, where $Z \sim \text{Beta}(2 + S\theta, 2 - S\theta)$ and $\theta = e_1 = (1, 0, \ldots, 0)$. The parameter $\theta$ is estimated by maximizing the log-likelihood.

In Swimmer, the action is two-dimensional, i.e., $A = (A_1, A_2)$, and we sample each component independently given the state: $A_1 \sim \text{Beta}(2 + S\theta_1, 2 - S\theta_1)$ and $A_2 \sim \text{Beta}(2 + S\theta_2, 2 - S\theta_2)$, with $\theta_1 = e_1 = (1, 0, \ldots, 0)$ and $\theta_2 = e_2 = (0, 1, 0, \ldots, 0)$.

The results, summarized in Figure 3, demonstrate that using history-dependent behavior policy estimation generally reduces the MSE of OIS in large-sample settings. Moreover, the performance tends to improve with longer history lengths.

We further evaluate the use of history-dependent behavior policies in the SIS, DR, and MIS estimators within the more complex Swimmer environment. Results, presented in Figure 4, again aligns with our theory.

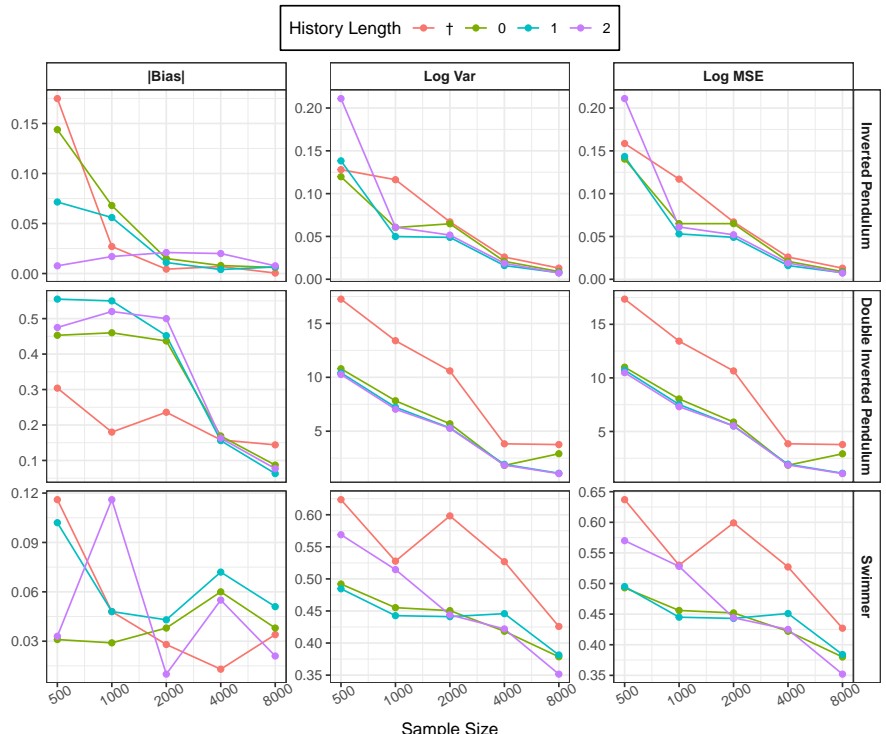

*Figure 3.* Bias, log variance and log MSE for OIS estimators across three different environments

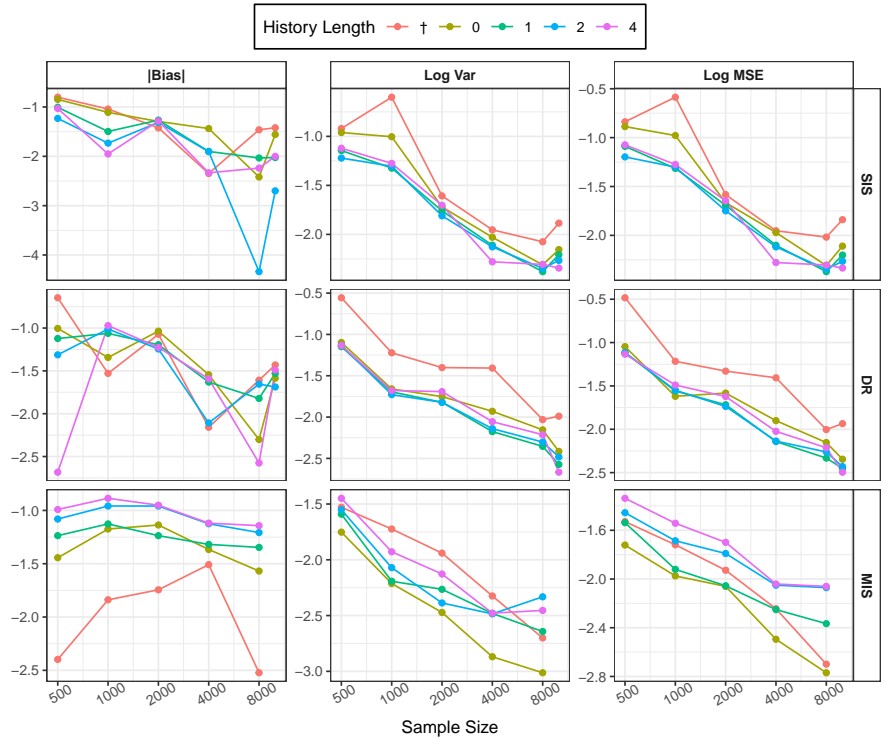

*Figure 4.* Bias, log variance and log MSE for OIS,DR and MIS estimators in Swimmer environment

## C. Proofs

### C.1. Proof of Lemma 1

According to the definitions of $\widehat{v}_{\mathrm{IS}}^{\mathrm{CD}}$ and $\widehat{v}_{\mathrm{IS}}^{\mathrm{CA}}$, it follows from straightforward calculations that

$$\widehat{v}_{\mathrm{IS}}^{\mathrm{CA}} = \mathbb{E}_n \Big\{ \sum_a \pi_e(a)\widehat{r}(a) + \frac{\pi_e(A)}{\widehat{\pi}_b(A)}[R - \widehat{r}(A)] \Big\},$$

and

$$\widehat{v}_{\mathrm{IS}}^{\mathrm{CD}} = \mathbb{E}_n \Big\{ \sum_a \pi_e(a)\widehat{r}(S, a) + \frac{\pi_e(A)}{\widehat{\pi}_b(A|S)}[R - \widehat{r}(S, A)] \Big\}.$$

According to Neyman orthogonality, both the estimated reward and estimated behavior policy can be asymptotically replaced by its oracle value without changing the OPE estimator's asymptotic MSE (Chernozhukov et al., 2018). As this part of the proof follows standard arguments, we provide only a sketch; interested readers may refer to, for example, the proof of Theorem 9 in Kallus & Uehara (2020) for further details.

Specifically, $\widehat{v}_{\mathrm{IS}}^{\mathrm{CD}}$ can be decomposed into the following four terms:

$$
\begin{aligned}
\widehat{v}_{\mathrm{IS}}^{\mathrm{CD}} &= \mathbb{E}_n \left( \sum_a \pi_e(a)r(S, a) + \frac{\pi_e(A)}{\pi_b(A)}[R - r(S, A)] \right) & (6) \\
&+ \mathbb{E}_n \left( \sum_a \pi_e(a|S)[\widehat{r}(S, a) - r(S, a)] - \frac{\pi_e(A)}{\pi_b(A)}[\widehat{r}(S, A) - r(S, A)] \right) & (7) \\
&+ \mathbb{E}_n \left[ \left( \frac{\pi_e(A)}{\widehat{\pi}_b(A|S)} - \frac{\pi_e(A)}{\pi_b(A)} \right)[R - r(S, A)] \right] & (8) \\
&+ \mathbb{E}_n \left( \frac{\pi_e(A)}{\widehat{\pi}_b(A|S)} - \frac{\pi_e(A)}{\pi_b(A)} \right)[\widehat{r}(S, A) - r(S, A)]. & (9)
\end{aligned}
$$

Here, the right-hand-side (RHS) of (6) is the oracle DR estimator with the true reward function and IS ratio, and (7) – (9) are the reminder terms, which we will show are of order $o_p(n^{-1/2})$. In particular:

- For fixed $\widehat{r}$ and $\widehat{\pi}_b$, (7) and (8) are of zero mean. They are of the order $o_p(n^{-1/2})$ provided that $\widehat{r}$ and $\widehat{\pi}_b$ converge to their oracle values. Even when $\widehat{r}$ and $\widehat{\pi}_b$ are estimated from the same data used in the evaluation, our use of tabular methods—combined with the fact that the number of contexts and actions is finite—ensures that these estimators belong to function classes with finite VC-dimension (Van Der Vaart et al., 1996). Therefore, standard empirical process theory (e.g., Chernozhukov et al., 2014, Corollary 5.1) can be applied to establish that these terms are indeed $o_p(n^{-1/2})$.

- For fixed $\widehat{r}$ and $\widehat{\pi}_b$, (9) is of the order $\|\widehat{r} - r\| \times \|\widehat{\pi}_b - \pi_b\|$ where $\|\widehat{r} - r\|$ and $\|\widehat{\pi}_b - \pi_b\|$ denote the root MSEs (RMSEs) between $\widehat{r}(S, A)$ and $r(S, A)$, and between $\widehat{\pi}_b(A|S)$ and $\pi_b(A)$, respectively. Crucially, the order is the product of the two RMSEs. Consequently, as they decay to zero at a rate of $o_p(n^{-1/4})$ – which is much slower than the parametric rate $O_p(n^{-1/2})$ – this term becomes $o_p(n^{-1/2})$ as well. Again, under tabular estimation with finitely many contexts and actions, these estimators converge at the parametric rate, and empirical process theories can be similarly used to handle the dependence between the estimators and the evaluation data in (9).

Therefore, $\widehat{v}_{\mathrm{IS}}^{\mathrm{CD}}$ is asymptotically equivalent to the oracle DR estimator (which is unbiased). Consequently, they achieve the same asymptotic variance and MSE, and we have

$$
\begin{aligned}
\mathrm{MSE}_A(\widehat{v}_{\mathrm{IS}}^{\mathrm{CD}}) &= \mathrm{MSE}_A \left[ \mathbb{E}_n \left( \sum_a \pi_e(a)r(S, a) + \frac{\pi_e(A)}{\pi_b(A)}[R - r(S, A)] \right) \right] \\
&= \mathrm{Var}_A \left[ \mathbb{E}_n \left( \sum_a \pi_e(a)r(S, a) + \frac{\pi_e(A)}{\pi_b(A)}[R - r(S, A)] \right) \right] \\
&= \frac{1}{n}\mathrm{Var} \left( \sum_a \pi_e(a)r(S, a) + \frac{\pi_e(A)}{\pi_b(A)}[R - r(S, A)] \right),
\end{aligned}
$$

which is equal to

$$\frac{1}{n}\text{Var}\left(\sum_a \pi_e(a)r(S,a)\right) + \frac{1}{n}\text{Var}\left(\frac{\pi_e(A)}{\pi_b(A)}[R - r(S,A)]\right).$$

Similar argument yields that

$$\text{MSE}_A(\widehat{v}_{\text{IS}}^{\text{CA}}) = \frac{1}{n}\text{Var}\left(\frac{\pi_e(A)}{\pi_b(A)}[R - \mathbb{E}(R|A)]\right).$$

Then the first inequality follows from the fact that

$$\text{Var}\left(\frac{\pi_e(A)}{\pi_b(A)}[R - \mathbb{E}(R|A)]\right) = \text{Var}\left(\frac{\pi_e(A)}{\pi_b(A)}[R - r(S,A)]\right) + \text{Var}\left(\frac{\pi_e(A)}{\pi_b(A)}[r(S,A) - \mathbb{E}(R|A)]\right),$$

and that

$$\text{Var}\left(\frac{\pi_e(A)}{\pi_b(A)}[r(S,A) - \mathbb{E}(R|A)]\right) \geq \text{Var}\left[\mathbb{E}\left(\frac{\pi_e(A)}{\pi_b(A)}[r(S,A) - \mathbb{E}(R|A)]|S\right)\right] = \text{Var}\left(\sum_a \pi_e(a)r(S,a)\right).$$

The equality holds if and only if $\text{Var}\left(\frac{\pi_e(A)}{\pi_b(A)}[r(S,A) - \mathbb{E}(R|A)]|S\right) = 0$, which implies that the context $S$ is independent of the reward function $r$.

We next prove the second inequality. Since $\widehat{v}_{\text{IS}}^{\dagger}$ is unbiased, the second inequality follows from the fact that

$$\begin{aligned}
\text{MSE}_A(\widehat{v}_{\text{IS}}^{\dagger}) &= \frac{1}{n}\text{Var}\left(\frac{\pi_e(A)}{\pi_b(A)}R\right) \\
&= \frac{1}{n}\text{Var}\left(\frac{\pi_e(A)}{\pi_b(A)}[R - \mathbb{E}(R|A)]\right) + \frac{1}{n}\text{Var}\left(\frac{\pi_e(A)}{\pi_b(A)}\mathbb{E}(R|A)\right). \\
&= \text{MSE}_A(\widehat{v}_{\text{IS}}^{\text{CA}}) + \frac{1}{n}\text{Var}\left(\frac{\pi_e(A)}{\pi_b(A)}\mathbb{E}(R|A)\right) \geq \text{MSE}_A(\widehat{v}_{\text{IS}}^{\text{CA}}).
\end{aligned}$$

The equality holds if and only if $\mathbb{E}(R|A) = 0$ almost surely.

### C.2. Proof of Theorems in Section 4

**Details of Assumption 1.** We assume that the policy class is parametrized by a vector $\theta = (\theta_0, \ldots, \theta_k)$. For any $\pi_\theta \in \Pi_k$ and $i \in \{0, \ldots, k\}$, the state-action pair $S_{t-i}, A_{t-i}$ affects $\theta$ only through their interactions with $\theta_i$. In this way, if we set $\theta_1 = \ldots = \theta_k = 0$, then $\pi_\theta$ becomes a Markov policy. Moreover, for any $k' < k$, if we fix $\theta_{k'+1} = \ldots = \theta_k = 0$, then the policy class $\Pi_k$ degenerates to $\Pi_{k'}$.

**Notations.** Given a single trajectory $H = (s_0, a_0, r_0, \ldots s_T, a_T, r_T)$, let $H_{t-k:t}$ denote the trajectory segment $(s_{t-k}, a_{t-k}, \ldots, s_t)$ the likelihood function of trajectory $H$ under policy $\pi_\theta(\cdot|\cdot)$ is given by

$$p(H, \theta) = \prod_{t=0}^{T} \pi_\theta(a_t|H_{t-k:t})p(r_t|s_t, a_t)p(s_{t+1}|s_t, a_t).$$

Further define $p(H, \pi_e)$ be the likelihood function of trajectory $H$ under policy $\pi_e$, given as

$$p(H, \pi_e) = \prod_{t=0}^{T} \pi_e(a_t|H_{t-k:t})p(r_t|s_t, a_t)p(s_{t+1}|s_t, a_t).$$

The loglikelihood function is defined as $L(H, \theta) = \log p(H, \theta)$ and the score function is defined as

$$s(H, k, \theta) = \frac{\partial}{\partial \theta}\log p(H, \theta) = \frac{\partial}{\partial \theta}\sum_{t=0}^{T}\log \pi_\theta(a_t|H_{t-k:t}).$$

In what follows, we write $s(H, k, \theta)$ as $s(H, \theta)$ to ease notation. Let $H_t = (s_0, a_0, \ldots, s_t, a_t)$ be the state-action trajectory up to time $t$ and $H_{s_t} = (s_0, a_0, \ldots, s_t)$ be the trajectory up to $s_t$. We further define

$$s(H_t, \theta) = \frac{\partial}{\partial\theta} \sum_{j=0}^{t} \log \pi_\theta(a_j | H_{j-k:j}),$$

$$s(H_{t:T}, \theta) = \frac{\partial}{\partial\theta} \sum_{j=t+1}^{T} \log \pi_\theta(a_j | H_{j-k:j}).$$

**Proof of Theorem 2.**
For simplicity of notation, we define

$$u(H, \theta) = G_T \prod_{t=0}^{T} \frac{\pi_e(a_t | s_t)}{\pi_\theta(a_t | H_{t-k:t})} = G_T \frac{p(H, \pi_e)}{p(H, \theta)}.$$

Direct calculation yields that

$$\frac{\partial}{\partial\theta} u(H, \theta) = -u(H, \theta) s(H, \theta), \tag{10}$$

and $\widehat{v}_{\mathrm{OIS}}^{\dagger} = \frac{1}{n} \sum_{i=1}^{n} u(H_i, \theta^*), \widehat{v}_{\mathrm{OIS}}(k) = \sum_{i=1}^{n} u(H_i, \widehat{\theta}_n)$. Using Taylor expansion at $\theta = \theta^*$, we obtain

$$
\begin{aligned}
\widehat{v}_{\mathrm{OIS}}(k) - \widehat{v}_{\mathrm{OIS}}^{\dagger} &= \frac{1}{n} \sum_{i=1}^{n} \frac{\partial}{\partial\theta} u(H_i, \theta^*)(\widehat{\theta}_n - \theta^*) + R_n(\widehat{\theta}_n) \\
&= \frac{1}{n} \sum_{i=1}^{n} u(H_i, \theta^*) s(H_i, \theta^*)(\widehat{\theta}_n - \theta^*) + R_{n1}(\widehat{\theta}_n), \tag{11}
\end{aligned}
$$

where the remainder term can be represented as

$$R_{n1}(H, \widehat{\theta}) = \frac{1}{2n} u(H, \widetilde{\theta}_n)(\widehat{\theta}_n - \theta^*)^{\top} \sum_{i=1}^{n} \left[ s(H, \widetilde{\theta}) s(H, \widetilde{\theta})^{\top} - \frac{\partial}{\partial\theta} s(H, \widetilde{\theta}) \right] (\widehat{\theta}_n - \theta^*).$$

Under the bounded rewards assumption (Assumption 3), we have $G_T = O(T R_{\max})$. Under the coverage assumption (Assumption 4, we have $u(H, \theta) = O_p(T C^T R_{\max})$ and $s(H, \theta) = O(\varepsilon^{-1})$. Under the differentiability assumption (Assumption 5), $\frac{\partial}{\partial\theta} s(H, \theta) = O(\varepsilon^{-2})$. Combining these facts, we obtain that the remainder term satisfies

$$R_{n1} = O_p \left( \frac{T C^T R_{\max}}{\varepsilon^2} \|\widehat{\theta}_n - \theta^*\|^2 \right). \tag{12}$$

Using the property of maximum likelihood estimator (see e.g., Theorem 4.17 in Shao, 2003), we have

$$\sqrt{n}(\widehat{\theta}_n - \theta^*) = I^{-1}(\theta^*) \frac{1}{\sqrt{n}} \sum_{i=1}^{n} s(H_i, \theta^*) + O_P(\|\widehat{\theta} - \theta^*\|^2). \tag{13}$$

Further using the central limit theorem, $\sqrt{1/nT} \sum_{i=1}^{n} s(H_i, \theta^*)$ converges to a normal distribution with mean zero and variance $I(\theta^*)$, which is of order $O(T)$. It follows that under the non-singularity assumption (Assumption 6), $\|\widehat{\theta}_n - \theta^*\| = O_P \left( \frac{k+1}{\sqrt{nT}} \right)$. Combining equations (11), (12) and (13), we have

$$
\begin{aligned}
\widehat{v}_{\mathrm{OIS}}(k) - \widehat{v}_{\mathrm{OIS}} &= -\frac{1}{n} \sum_{i=1}^{n} u(H_i, \theta^*) s(H_i, \theta^*) I^{-1}(\theta^*) \frac{1}{n} \sum_{j=1}^{n} s(H_j, \theta^*) + O_p \left( \frac{(k+1)C^T R_{\max}}{n\varepsilon^2} \right) \\
&= -\frac{1}{\sqrt{n}} \mathbb{E}[u(H, \theta^*) s(H, \theta^*)] I^{-1}(\theta^*) \frac{1}{\sqrt{n}} \sum_{j=1}^{n} s(H_j, \theta^*) + O_p \left( \frac{(k+1)C^T R_{\max}}{n\varepsilon^2} \right) + R_{n2}, \tag{14}
\end{aligned}
$$

where

$$R_{n2} = \frac{1}{n}\left\{\frac{1}{n}\sum_{i=1}^{n}u(H_i,\theta^*)s(H_i,\theta^*) - \mathbb{E}[u(H,\theta^*)s(H,\theta^*)]\right\}I^{-1}(\theta^*)\frac{1}{n}\sum_{j=1}^{n}s(H_j,\theta^*).$$

Again, according to the central limit theorem, we have

$$\frac{1}{n}\sum_{i=1}^{n}u(H_i,\theta^*)s(H_i,\theta^*) - \mathbb{E}[u(H,\theta^*)s(H,\theta^*)] = O_p\left(\sqrt{\frac{T}{n}}C^T R_{\max}\varepsilon^{-1}\right).$$

Therefore, we obtain $R_{n2}$ is also of order $O_p\left(\frac{(k+1)C^T R_{\max}}{n\varepsilon^2}\right)$. Plug into equation (14), we obtain

$$\widehat{v}_{\text{OIS}}(k) - \widehat{v}_{\text{OIS}}^{\dagger} = -\frac{1}{\sqrt{n}}\mathbb{E}[u(H,\theta^*)s(H,\theta^*)]I^{-1}(\theta^*)\frac{1}{\sqrt{n}}\sum_{j=1}^{n}s(H_j,\theta^*) + O_p\left(\frac{(k+1)C^T R_{\max}}{n\varepsilon^2}\right),$$

where the predominant term on the right hand side is denoted as $v_1$. Using the fact that $\mathbb{E}[s(H,\theta^*)] = 0$, we know that the predominant term has mean 0. Meanwhile, since $I(\theta^*) = \mathbb{E}[s(H,\theta^*)s^{\top}(H,\theta^*)]$, we obtain

$$\text{Var}(v_1) = \text{Cov}(v_{\text{OIS}}, v_1) = \frac{1}{n}\mathbb{E}[u(H,\theta^*)s^{\top}(H,\theta^*)]I^{-1}(\theta^*)\mathbb{E}[u(H,\theta^*)s(H,\theta^*)]. \quad (15)$$

It follows that $\text{Cov}(v_{\text{OIS}}^{\dagger} - v_1, v_1) = 0$. We define

$$\mathbb{T}^{\perp}(k) := \left\{w = s^{\top}(H,\theta^*)a | a \in \mathbb{R}^k\right\}$$

as the tangent space spanned by score vector, and we define

$$\mathbb{T}(k) := \left\{w | \mathbb{E}\left\{u \cdot w\right\} = 0, \forall u \in \mathbb{T}^{\perp}(k)\right\}.$$

In fact, the whole space $\mathbb{R}^k$ can be decomposed into $\mathbb{T}(k)\bigoplus \mathbb{T}(k)^{\perp}$. $v_1 \in \mathbb{T}(k)^{\perp}$ is the orthogonal projection of $v_{\text{OIS}}^{\dagger}$ onto the tangent space spanned by the score vector and $v_{\text{OIS}}^{\dagger} - v_1 \in \mathbb{T}(k)$ is the projection of $v_{\text{OIS}}^{\dagger}$ on the space of random vectors orthogonal to the score vector. Moreover, equation (15) indicates

$$\widehat{v}_{\text{OIS}}(k) - v_{\text{true}} = (\widehat{v}_{\text{OIS}}^{\dagger} - v_{\text{true}}) - v_1 + R_{n3}, \quad (16)$$

with $R_{n3} = O_p\left(\frac{(k+1)C^T R_{\max}}{n\varepsilon^2}\right)$. Take variance on both sides, we obtain

$$\text{Var}(\widehat{v}_{\text{OIS}}(k)) = \text{Var}(\widehat{v}_{\text{OIS}}^{\dagger} - v_1) + \text{Var}(R_{n3}) + 2\text{Cov}(\widehat{v}_{\text{OIS}}^{\dagger} - v_1, R_{n3}). \quad (17)$$

Using similar calculations, we can show that

$$\text{Var}(\widehat{v}_{\text{OIS}}^{\dagger} - v_1) = O(R_{\max}^2 C^{2T}/n),$$
$$\text{Var}(R_{n3}) = O\left(\frac{(k+1)^2 C^{2T} R_{\max}^2}{n^2\varepsilon^4}\right).$$

By Cauchy-Schwartz inequality, we have

$$\text{Cov}(\widehat{v}_{\text{OIS}}^{\dagger} - v_1, R_{n3}) \le \sqrt{\text{Var}(\widehat{v}_{\text{OIS}}^{\dagger} - v_1) \cdot \text{Var}(R_{n3})} = O\left(\frac{(k+1)C^{2T} R_{\max}^2}{n^{3/2}\varepsilon^2}\right).$$

Since $\varepsilon$ is a constant, $\text{Var}(R_{n3})$ is a higher order term compared to $\text{Cov}(\widehat{v}_{\text{OIS}}^{\dagger} - v_1, R_{n3})$. Furthermore, since $\widehat{v}_{\text{OIS}}^{\dagger}$ is unbiased, so

$$\text{Bias}(\widehat{v}_{\text{OIS}}(k)) = O\left(\frac{(k+1)C^T R_{\max}}{n\varepsilon^2}\right).$$

It follows that $\text{Bias}^2(\widehat{v}_{\text{OIS}}(k))$ is a higher order term compared to $\text{Cov}(\widehat{v}_{\text{OIS}}^\dagger - v_1, R_{n3})$. Using bias-variance decomposition, we obtain

$$
\begin{aligned}
\text{MSE}(\widehat{v}_{\text{OIS}}(k)) &= \text{Var}(\widehat{v}_{\text{OIS}}^\dagger - v_1) + \text{Bias}^2(\widehat{v}_{\text{OIS}}(k)) + O\left(\frac{(k+1)C^{2T}R_{\max}^2}{n^{3/2}\varepsilon^2}\right) \\
&= \text{Var}\left(\text{Proj}_{\mathbb{T}(k)}(\widehat{v}_{\text{OIS}}^\dagger)\right) + O\left(\frac{(k+1)C^{2T}R_{\max}^2}{n^{3/2}\varepsilon^2}\right) \\
&= \frac{1}{n}\left(\text{Proj}_{\mathbb{T}(k)}(\lambda_T G_T)\right) + O\left(\frac{(k+1)C^{2T}R_{\max}^2}{n^{3/2}\varepsilon^2}\right),
\end{aligned}
\tag{18}
$$

where $\text{Proj}_{\mathbb{T}(k)}(\cdot)$ represents the orthogonal projection of a random variable to the space $\mathbb{T}(k)$. This proves the first claim of Theorem 2.

We next show the second claim of Theorem 2. In fact, for any $k' < k$, under the monotocity assumption (Assumption 1), the tangent space spanned by score vector for model $\Pi_k$ is strictly larger than that of $\Pi_{k'}$. Therefore, we have $\mathbb{T}(k)^\perp \subseteq \mathbb{T}(k')^\perp$. It follows that $k' < k$, $\mathbb{T}(k') \subseteq \mathbb{T}(k)$ and the second claim of Theorem 2 directly follows from Pythagorean Theorem.

**Proof of Theorem 4.**
The proof of Theorem 4 simply follows the proof of Theorem 6 by taking $Q(s,a) \equiv 0$ and is thus omitted.

**Proof of Theorem 6.**
The likelihood of trajectory segment $H_t = (S_0, A_0, R_0, \ldots, S_t, A_t, R_t)$ can be represented as:

$$
P_\theta(H_{S_{t+1}}) = \prod_{j=0}^{t} \pi_\theta(A_j|H_{j-k:j})p(S_{j+1}|S_j, A_j)p(R_j|S_j, A_j).
$$

It follows that the cumulative density ratio with respect to behavior policy $\pi_\theta$ can be represented as

$$
\lambda_t(\theta) := \prod_{j=1}^{t} \frac{\pi_e(A_j|S_j)}{\pi_\theta(A_j|S_{j-k:j})} = \frac{P_{\pi_e}(H_{S_{t+1}})}{P_\theta(H_{S_{t+1}})}.
$$

Then the doubly robust estimator can be represented as

$$
\begin{aligned}
\widehat{v}_{\text{DR}}(k) &= \mathbb{E}_n \sum_{t=0}^{T} \left\{ \frac{P_{\pi_e}(H_{S_{t+1}})}{P_{\widehat{\theta}_n}(H_{S_{t+1}})}\gamma^t(R_t - Q_t(S_t, A_t)) + \frac{P_{\pi_e}(H_{S_t})}{P_{\widehat{\theta}_n}(H_{S_t})}\gamma^t Q_t(S_t, \pi_e) \right\} \\
&= \mathbb{E}_n Q_0(S_0, \pi_e) + \mathbb{E}_n \sum_{t=0}^{T} \left\{ \frac{P_{\pi_e}(H_{S_{t+1}})}{P_{\widehat{\theta}_n}(H_{S_{t+1}})}\gamma^t(R_t - Q_t(S_t, A_t) + \gamma Q_{t+1}(S_{t+1}, \pi_e)) \right\},
\end{aligned}
\tag{19}
$$

with $Q_t(S, \pi_e) = \int_a Q_t(S,a)d\pi_e(a|S)$ and the doubly robust estimator with oracle weight can be represented as

$$
\widehat{v}_{\text{DR}}^\dagger = \mathbb{E}_n Q_0(S_0, \pi_e) + \mathbb{E}_n \sum_{t=0}^{T} \left\{ \frac{P_{\pi_e}(H_{S_{t+1}})}{P_{\theta^*}(H_{S_{t+1}})}\gamma^t(R_t - Q_t(S_t, A_t) + \gamma Q_{t+1}(S_{t+1}, \pi_e)) \right\}.
$$

For notation simplicity, we denote

$$
u(H_{S_{t+1}}, \theta) = \frac{P_{\pi_e}(H_{A_t})}{P_{\pi_\theta}(H_{A_t})}\gamma^t(R_t - Q_t(S_t, A_t) + \gamma Q_{t+1}(S_{t+1}, \pi_e)).
$$

Then direct calculation yields that $\frac{\partial}{\partial\theta}u(H_{S_{t+1}}, \theta) = u(H_{S_{t+1}}, \theta)s(H_t, \theta)$. Under Assumption 3, 4, 5, using similar argument as proving equation (11), (12),(13) and (14), Taylor expansion yields

$$
\begin{aligned}
\widehat{v}_{\text{DR}}(k) - \widehat{v}_{\text{DR}}^\dagger &= -\mathbb{E}_n\left\{\sum_{t=0}^{T} u(H_{S_{t+1}})s(\theta^*, H_{A_t})^T\right\}(\widehat{\theta}_n - \theta^*) + O_P\left(\frac{(k+1)TC^T U_{\max}}{\varepsilon^2}\|\widehat{\theta}_n - \theta^*\|^2\right) \\
&= -\mathbb{E}_n\left\{\sum_{t=0}^{T} u(H_{S_{t+1}})s(\theta^*, H_{A_t})^T\right\}I^{-1}(\theta^*)\mathbb{E}_n s(\theta^*, H_T) + O_P\left(\frac{(k+1)C^T U_{\max}}{n\varepsilon^2}\right) \\
&= -\mathbb{E}\left\{\sum_{t=0}^{T} u(H_{S_{t+1}})s(\theta^*, H_{A_t})^T\right\}I^{-1}(\theta^*)\mathbb{E}_n s(\theta^*, H_T) + O_P\left(\frac{(k+1)C^T U_{\max}}{n\varepsilon^2}\right).
\end{aligned}
$$

Denote the main term on the right hand side on the last line by $v_2$. Noted that

$$
\begin{aligned}
\mathbb{E}\left\{\sum_{t=0}^{T} u(H_{S_{t+1}}, \theta^*) s(H_T, \theta^*)\right\} &= \mathbb{E}\left\{\sum_{t=0}^{T} u(H_{S_{t+1}}, \theta^*)\left(s(H_t, \theta^*) + s(H_{t:T}, \theta^*)\right)\right\} \\
&= \mathbb{E}\left\{\mathbb{E}\left[\sum_{t=0}^{T} u(H_{S_{t+1}}, \theta^*)\left(s(H_t, \theta^*) + s(H_{t:T}, \theta^*)\right)\bigg| H_{S_{t+1}}\right]\right\} \\
&= \mathbb{E}\left\{\sum_{t=0}^{T} u(H_{S_{t+1}}, \theta^*)\left(s(H_t, \theta^*) + \mathbb{E}\left[s(H_{t:T}, \theta^*)\big| H_{S_{t+1}}\right]\right)\right\} \\
&= \mathbb{E}\left\{\sum_{k=0}^{T} u(H_{S_{t+1}}, \theta^*)\left(s(H_t, \theta^*) + \mathbb{E}\left[s(H_{t:T}, \theta^*)\big| S_{t+1}\right]\right)\right\} \\
&= \mathbb{E}\left\{\sum_{k=0}^{T} u(H_{S_{t+1}}, \theta^*) s(H_t, \theta^*)\right\},
\end{aligned}
$$

where the second equality follows from total expectation formula, the fourth equality follows from the Markov property and the last equality follows from the fact that the score function vanishes at the true parameter. Thus, it follows from direct calculation that $\mathrm{Var}(v_2) = \mathrm{Cov}(\widehat{v}_{\mathrm{DR}}^\dagger, v_2)$. Therefore, similar to the proof of Theorem 2, we know that $v_2$ is the orthogonal projection of $\widehat{v}_{\mathrm{DR}}^\dagger$ onto the tangent space spanned by score function. Plugging into equation (20) and minus $v_{\mathrm{true}}$ on both sides yields

$$
\widehat{v}_{\mathrm{DR}}(k) - v_{\mathrm{true}} = (\widehat{v}_{\mathrm{DR}}^\dagger - v_{\mathrm{true}}) - v_2 + O_P\left(\frac{(k+1)C^T U_{\max}}{n\varepsilon^2}\right).
$$

Using similar argument as proving equation (18) and combining the fact that $\widehat{v}_{\mathrm{DR}}^\dagger$ is unbiased and $\mathbb{E}v_2 = 0$, we obtain

$$
\mathrm{MSE}(\widehat{v}_{\mathrm{DR}}(k)) = \mathrm{Var}(\mathrm{Proj}_{\mathbb{T}(k)}(\widehat{v}_{\mathrm{DR}}^\dagger)) + O\left(\frac{(k+1)C^{2T} U_{\max}^2}{n^{3/2}\varepsilon^2}\right). \tag{20}
$$

This finishes the first claim of Theorem 6. In order to prove $\mathrm{Var}(\mathrm{Proj}_{\mathbb{T}(k)}(\widehat{v}_{\mathrm{DR}}^\dagger))$ is decreasing with respect to $k$, we denote $\sigma^2(k) = \mathrm{Var}(\widehat{v}_{\mathrm{DR}}^\dagger) - \mathrm{Var}(\mathrm{Proj}_{\mathbb{T}(k)}(\widehat{v}_{\mathrm{DR}}^\dagger))$, then $\sigma^2(k) = \mathrm{Var}(v_2)$. It follows that

$$
\begin{aligned}
\sigma^2(k) =&\frac{1}{n}\mathbb{E}\left\{\sum_{t=0}^{T} u(H_{S_{t+1}}, \theta^*) s^\top(H_t, \theta^*)\right\} I^{-1}(\theta^*)\mathbb{E}\left\{\sum_{t=0}^{T} u(H_{S_{t+1}}, \theta^*) s(H_t, \theta^*)\right\} \\
=&\frac{1}{n}\mathbb{E}\left\{\sum_{t=0}^{T}\prod_{j=0}^{t}\frac{\pi_e(A_j|S_j)}{\pi_{\theta^*}(A_j|S_j)}\gamma^t U_t s(H_t, \theta^*)^T\right\} I^{-1}(\theta^*)\mathbb{E}\left\{\sum_{t=0}^{T}\prod_{j=0}^{t}\frac{\pi_e(A_j|S_j)}{\pi_{\theta^*}(A_j|S_j)}\gamma^t U_t s(H_t, \theta^*)\right\}. \tag{21}
\end{aligned}
$$

with

$$
U_t = R_t - Q_t(S_t, A_t) + \gamma Q_{t+1}(S_{t+1}, \pi_e).
$$

We next prove that for any $k' < k$, the inequality $\sigma^2(k') \leq \sigma^2(k)$ holds. For $\theta = (\theta_0, \ldots, \theta_k)$, define $\gamma = (\theta_0, \ldots, \theta_{k'})$, $\eta = (\theta_{k'+1}, \ldots, \theta_k)$ and $\theta^* = (\gamma^*, \eta^*)$. It follows that $s^\top(H_t, \theta) = (s^\top(H_t, \gamma), s^\top(H_t, \eta))$ for any $t \in \{0, 1, \ldots, T\}$. Therefore, we can conclude that

$$
\sigma^2(k') = \frac{1}{n}\mathbb{E}\left\{\sum_{t=0}^{T}\prod_{j=0}^{t}\frac{\pi_e(A_j|S_j)}{\pi_{\theta^*}(A_j|S_j)}U_t s(H_t, \gamma^*)^\top\right\} I^{-1}(\gamma^*)\mathbb{E}\left\{\sum_{t=0}^{T}\prod_{j=0}^{t}\frac{\pi_e(A_j|S_j)}{\pi_{\theta^*}(A_j|S_j)}U_t s(H_t, \gamma^*)\right\}.
$$

Let $I(\gamma^*) = \mathbb{E}[s(H, \gamma^*)s^\top(H, \gamma^*)]$, $I(\eta^*) = \mathbb{E}[s(H, \eta^*)s^\top(H, \eta^*)]$ and $I_{12} = \mathbb{E}[s(H, \gamma^*)s^\top(H, \eta^*)]$, then

$$
I(\theta^*) = \begin{bmatrix} I(\gamma^*) & I_{12} \\ I_{12}^T & I(\eta^*) \end{bmatrix},
$$

In order to calculate $I^{-1}(\theta^*)$, we apply the formula of the inversion of a block matrix:

$$\begin{bmatrix} A & B \\ C & D \end{bmatrix}^{-1} = \begin{bmatrix} A^{-1} + A^{-1}B(D - CA^{-1}B)^{-1}CA^{-1} & -A^{-1}B(D - CA^{-1}B)^{-1} \\ -(D - CA^{-1}B)^{-1}CA^{-1} & (D - CA^{-1}B)^{-1} \end{bmatrix},$$

we obtain from equation (21) that

$$
\begin{aligned}
\sigma^2(k) &= \sigma^2(k') + \mathbb{E}\left[\sum_{t=0}^{T} u(H_{S_{t+1}})s^\top(H_T, \gamma^*)\right] I^{-1}(\gamma^*) I_{12} J^{-1} I_{21} I^{-1}(\gamma^*) \mathbb{E}\left[\sum_{t=0}^{T} u(H_{S_{t+1}})s(H_T, \gamma^*)\right] \\
&\quad + \mathbb{E}\left[\sum_{t=0}^{T} u(H_{S_{t+1}})s^\top(H_T, \eta^*)\right] J^{-1} I_{12}^T I^{-1}(\gamma^*) \mathbb{E}\left[\sum_{t=0}^{T} u(H_{S_{t+1}})s(H_T, \gamma^*)\right] \\
&\quad + \mathbb{E}\left[\sum_{t=0}^{T} u(H_{S_{t+1}})s^\top(H_T, \gamma^*)\right] I^{-1}(\gamma^*) I_{12} J^{-1} \mathbb{E}\left[\sum_{t=0}^{T} u(H_{S_{t+1}})s(H_T, \eta^*)\right] \\
&\quad - \mathbb{E}\left[\sum_{t=0}^{T} u(H_{S_{t+1}})s^\top(H_T, \eta^*)\right] J^{-1} \mathbb{E}\left[\sum_{t=0}^{T} u(H_{S_{t+1}})s(H_T, \eta^*)\right] \\
&= \sigma^2(k') + \mathbb{E}\left\| J^{-1/2} I_{12}^T I^{-1}(\gamma^*) \sum_{t=0}^{T} u(H_{S_{t+1}})s(H_t, \gamma^*) - J^{-1/2} \sum_{t=0}^{T} u(H_{S_{t+1}})s(H_t, \eta^*) \right\|^2,
\end{aligned}
$$

with $J = I(\eta^*) - I_{12}^T I^{-1}(\gamma^*) I_{12}$. Thus, we obtain $\sigma^2(k) \geq \sigma^2(k')$ for any $k' < k$. To this end, we finishes the proof of $\mathrm{Var}(\mathrm{Proj}_{\mathbb{T}(k)}(\widehat{v}_{\mathrm{DR}}^\dagger))$ is decreasing with respect to $k$.

**Proof of Corollary 7.**

We directly calculate $\sigma^2(k)$ in equation (21).

$$
\begin{aligned}
\sigma^2(k) &= \mathbb{E}\left\{ \prod_{j=0}^{t} \frac{\pi_e(A_j|S_j)}{\pi_{\theta^*}(A_j|S_j)} \gamma^t U_t s(\theta^*, H_t) \right\} \\
&= \mathbb{E}\left\{ \mathbb{E}\left[ \prod_{j=0}^{t} \frac{\pi_e(A_j|S_j)}{\pi_{\theta^*}(A_j|S_j)} \gamma^t U_t s(\theta^*, H_t) \Big| H_t \right] \right\} \\
&= \mathbb{E}\left\{ \prod_{j=0}^{t} \frac{\pi_e(A_j|S_j)}{\pi_{\theta^*}(A_j|S_j)} s(\theta^*, H_t) \gamma^t \mathbb{E}\left[ U_t | S_t, A_t \right] \right\} \\
&= 0, \qquad\qquad (22)
\end{aligned}
$$

where the last equality follows from Bellman equation, which indicates $\mathbb{E}\left[ U_t | S_t, A_t \right] = 0$. Together with equation (21) completed the proof.

**Proof of Theorem 8.**

We first prove that the MIS estimators with weight function estimated by linear sieves is equivalent to the double reinforcement learning (DRL) estimator (Kallus & Uehara, 2020) with $Q$-function estimated by linear sieve, that is

$$\widehat{v}_{\mathrm{MIS}} = \widehat{v}_{\mathrm{DRL}} := \mathbb{E}_n\left\{ \sum_{t=0}^{T} \widehat{w}_t \left( R_t - \widehat{Q}_t(S_t, A_t) \right) + \widehat{w}_{t-1} \sum_a \widehat{Q}_t(S_t, a)\pi_e(a|S_t) \right\},$$

where $\widehat{Q}_t = \mathbb{E}_n \phi_t^\top(A_t, S_t)\widehat{\beta}_t$, and $\beta_t$ is iteratively defined as $\widehat{\beta}_t = \widehat{\Sigma}_t^{-1}(\mathbb{E}_n R_t + \gamma \widehat{\Sigma}_{t+1,t}\widehat{\beta}_{t+1})$.

For ease of notation, we define

$$
\begin{aligned}
\widehat{Q}_t(S, \pi_e) &:= \sum_a \widehat{Q}_t(S, a)\pi_e(a|S), \\
\phi_t(S, \pi_e) &:= \sum_a \phi_t(S, a)\pi_e(a|S).
\end{aligned}
$$

Recall that $\widehat{w}_t = \phi_t(S_t, A_t)\widehat{\alpha}_t$. By direct calculation, we have

$$
\begin{aligned}
\mathbb{E}_n\left\{\widehat{w}_{t-1}\widehat{Q}_t(S_t, \pi_e)\right\} &= \mathbb{E}_n\left\{\widehat{\alpha}_{t-1}^\top \phi_{t-1}(S_t, A_t)\phi_t^\top(S_t, \pi_e)\widehat{\beta}_t\right\} = \widehat{\alpha}_{t-1}^\top \widehat{\Sigma}_{t,t-1}\widehat{\beta}_t. \\
\mathbb{E}_n\left\{\widehat{w}_t\widehat{Q}_t(S_t, A_t)\right\} &= \mathbb{E}_n\left\{\widehat{\alpha}_t^\top \phi_t(S_t, A_t)\phi_t(S_t, A_t)^\top\widehat{\beta}_t\right\} \\
&= \mathbb{E}_n\left\{(\widehat{\Sigma}_t^{-1}\widehat{\Sigma}_{t,t-1}\widehat{\alpha}_{t-1})^\top \phi_t\phi_t^\top\widehat{\beta}_t\right\} \\
&= \widehat{\alpha}_{t-1}^\top \widehat{\Sigma}_{t,t-1}\widehat{\beta}_t.
\end{aligned}
\tag{23}
$$

where the second to last equality is obtained by the recursive definition of $\alpha_t$. It follows that $\mathbb{E}_n\widehat{w}_{t-1}\widehat{Q}_t(S_t, \pi_e) = \mathbb{E}_n\widehat{w}_t\widehat{Q}_t(S_t, A_t)$. Plugging into equation (23), we know that the MIS estimator is equivalent to DRL estimator.

Now, suppose the estimated weight and $Q$-function converges to its true value, then if we replace $\widehat{w}(S_t, A_t)$ by $\widehat{w}(S_{t:t-k}, A_{t:t-k})$, the resulting estimator will have a larger variance. Additionally, if the weight is estimated using all the history data, then $\widehat{v}_{\mathrm{MIS}}$ becomes the doubly robust $\widehat{v}_{\mathrm{DR}}$ estimator. The following theorem formalizes this result, indicating that for DRL estimator, the variance increases as more history are used to estimate the weights:

$$
\widehat{v}_{\mathrm{MIS}}(k) = \mathbb{E}_n\left\{\sum_{t=0}^{T} \widehat{w}_t(H_{t-k:t})\left(R_t - \widehat{Q}(S_t, A_t)\right) + \widehat{w}_{t-1}(H_{t-k-1:t-1})\int_a \widehat{Q}(S_t, a)d\pi_e(a|S_t)\right\}.
$$

We further assume that $\|\widehat{w}_t - w_t\| = o_P(n^{-1/4})$ and $\|\widehat{Q}_t - Q_t\| = o_P(n^{-1/4})$, where $\|\widehat{w}_t - w_t\|$ and $\|\widehat{Q}_t - Q_t\|$ denote the root MSEs (RMSEs) between $\widehat{w}_t(H_{t-k:t})$ and $w(H_{t-k:t})$, and between $\widehat{Q}_t(S, A)$ and $Q_t(S, A)$, According to Neyman orthogonality, both the estimated reward and estimated behavior policy can be asymptotically replaced by its oracle value (Chernozhukov et al., 2018) without changing the OPE estimator's asymptotic MSE (see also equations (6) - (9) for detailed explanation).

Therefore, we obtain that

$$
\widehat{v}_{\mathrm{MIS}}(k) = \mathbb{E}_n\left\{\sum_{t=0}^{T} w_t(R_t - Q_t(S_t, A_t)) + w_{t-1}Q_t(S_t, \pi_e)\right\} + o_P(n^{-1/2}).
$$

After rearranging the predominant term, we obtain that $\widehat{v}_{\mathrm{MIS}}(k)$ is asymptotically equals to

$$
\widehat{v}_{\mathrm{MIS}}(k) = \mathbb{E}_n Q_0(S_0, \pi_e) + \mathbb{E}_n\sum_{t=0}^{T} w_t(A_{t:t-k}, S_{t:t-k})(R_t - Q_t(S_t, A_t) + Q_{t+1}(S_{t+1}, \pi_e))
$$

If the $Q$ function is correctly specified, then

$$
\begin{aligned}
&\mathbb{E}\left[w_t(A_{t-k:t}, S_{t-k:t})\left(R_t - Q_t(S_t, A_t) + Q_{t+1}(S_{t+1}, \pi_e)\right)|S_{t-k:t}, A_{t-k:t}\right] \\
=\ & w_t(A_{t-k:t}, S_{t-k:t})\mathbb{E}\left[R_t - Q_t(S_t, A_t) + Q_{t+1}(S_{t+1}, \pi_e)|S_{t-k:t}, A_{t-k:t}\right] \\
=\ & 0.
\end{aligned}
\tag{24}
$$

Denote $U_t = R_t - Q^{\pi_e}(S_t, A_t) + V^{\pi_e}(S_{t+1})$. Then for any $t' < t$,

$$
\begin{aligned}
&\mathrm{Cov}\left(w_t(A_{t-k:t}, S_{t-k:t})U_t, w_t(A_{t'-k:t'}, S_{t'-k:t'})U_{t'}\right) \\
=\ & \mathbb{E}\left[w_t(A_{t-k:t}, S_{t-k:t})U_t w_{t'}(A_{t'-k:t'}, S_{t'-k:t'})U_{t'}|S_{t-k:t}, A_{t-k:t}\right] \\
=\ & \mathbb{E}\left\{w_t(A_{t-k:t}, S_{t-k:t})w_{t'}(A_{t'-k:t'}, S_{t'-k:t'})U_{t'}\mathbb{E}[U_t|S_{t-k:t}, A_{t-k:t}]\right\} \\
=\ & 0.
\end{aligned}
\tag{25}
$$

It follows that,

$$
\begin{aligned}
\mathrm{Var}_A(\widehat{v}_{\mathrm{MIS}}(k)) &= \frac{1}{n}\mathrm{Var}\left(Q_0(S_0,\pi_e) + \sum_{t=0}^{T} w_t(A_{t-k:t}, S_{t-k:t})U_t\right) \\
&= \frac{1}{n}\mathrm{Var}(Q_0(S_0,\pi_e)) + \sum_{t=0}^{T}\mathrm{Var}\left(w_t(A_{t-k:t}, S_{t-k:t})U_t\right) \\
&= \frac{1}{n}\mathrm{Var}(Q_0(S_0,\pi_e)) + \frac{1}{n}\sum_{t=0}^{T}\mathrm{Var}\left(w_t(A_{t-k:t}, S_{t-k:t})\mathbb{E}\left[U_t|A_{t-k:t}, S_{t-k:t}\right]\right) \\
&\quad + \frac{1}{n}\sum_{t=0}^{T}\mathbb{E}\left(w_t^2(A_{t-k:t}, S_{t-k:t})\mathrm{Var}\left[U_t|A_{t-k:t}, S_{t-k:t}\right]\right) \\
&= \frac{1}{n}\mathrm{Var}(Q_0(S_0,\pi_e)) + \frac{1}{n}\sum_{t=0}^{T}E\left(w_t^2(A_{t-k:t}, S_{t-k:t})\sigma^2(A_t, S_t)\right),
\end{aligned}
\tag{26}
$$

where $\sigma^2(A_t, S_t) = \mathrm{Var}(U_t|A_{t-k:t}, S_{t-k:t})$. Therefore, for any $k' < k$,

$$
\begin{aligned}
&\mathbb{E}\left(w_t^2(A_{t-k':t}, S_{t-k':t})\sigma^2(A_t, S_t)\right) \\
&= \mathbb{E}\left\{\left(\mathbb{E}[w_t(A_{t-k:t}, S_{t-k:t})|A_{t-k':t}, S_{t-k':t}]\right)^2\sigma^2(A_t, S_t)\right\} \\
&\leq \mathbb{E}\left\{\mathbb{E}[w_t^2(A_{t-k:t}, S_{t-k:t})|A_{t-k':t}, S_{t-k':t}]\sigma^2(A_t, S_t)\right\} \\
&= \mathbb{E}\left(w_t^2(A_{t-k:t}, S_{t-k:t})\sigma^2(A_t, S_t)\right),
\end{aligned}
\tag{27}
$$

where the first equality is based on the fact that $w_t(A_{t-k':t}, S_{t-k':t}) = \mathbb{E}[\lambda_t|A_{t-k':t}, S_{t-k':t}] = \mathbb{E}\left\{\mathbb{E}[\lambda_t|A_{t-k:t}, S_{t-k:t}]\big|A_{t-k':t}, S_{t-k':t}\right\} = \mathbb{E}[w(A_{t-k:t}, S_{t-k:t})|A_{t-k':t}, S_{t-k':t}]$ and the second equality is based on Jensen's inequality. Thus, combining equations (26) and (27), we obtain that $\mathrm{Var}_A(\widehat{v}_{\mathrm{MIS}}(k')) \leq \mathrm{Var}_A(\widehat{v}_{\mathrm{MIS}}(k))$.

### C.3. Assumptions and proof of Theorem 9

**Regularity conditions for Theorem 9.**
We first introduce regularity conditions for Theorem 9. Suppose $\Theta$ is the parametric space equipped with a norm $\|\cdot\|$ ($\Theta$ is not necessarily finite-dimensional). Denote $\mathcal{H}$ be the set of all possible trajectories and $\theta_0$ be the true prameter. For trajectory $H$, let $L(H, \theta)$ be the log likelihood function. let $s(H, \theta)[\cdot]$ be the Fréchet derivative of $L(H, \theta)$ with respect to $\theta$. For any $h \in \Theta$, $s(H, \theta)[h]$ is defined by

$$
s(H, \theta)[h] = \frac{\partial}{\partial\eta}L(H, \theta + \eta h)\Big|_{\eta=0}.
$$

Let $\mathbb{P}$ be the probability measure of $H$ induced by behavior policy $\pi_{\theta_0}$ and $\mathbb{P}_n$ be the corresponding empirical probability measure. We impose the following regularity conditions.

**Assumption 8.** For any $\theta$ in a neighbourhood of $\theta_0$, $\mathbb{P}\left\{s(H, \theta) - s(H, \theta_0)\right\} = O(\|\theta - \theta_0\|)$.

**Assumption 9.** For any $\theta \in \Theta$, there exists a corresponding $\theta_{0n}$ in the sieve space $\Theta_n$, such that $\|\theta - \theta_{0n}\| = o(n^{-1/4})$.

**Assumption 10.** $\theta_n$ is a consistent estimator of $\theta_0$ with $\|\theta_n - \theta_0\| = o_P(n^{-1/4})$.

**Assumption 11.** For some $\delta > 0$, the function class $\mathcal{F}_\delta = \{s(H, \theta) - s(H, \theta_0) : \|\theta - \theta_0\| < \delta, H \in \mathcal{H}\}$ is a $\mathbb{P}$-Donsker class.

**Assumption 12.** $s(H, \theta)[h]$ is Fréchet differentiable at the true parameter $\theta_0$ with a continuous derivative $\dot{s}_{\theta_0}[\cdot, h]$ which satisfies

$$
\mathbb{P}\left\{s(H, \hat{\theta}_n)[h] - s(H, \theta_0)[h] - \dot{s}_{\theta_0}[\hat{\theta}_n - \theta_0, h]\right\} = o_P(n^{-1/2}).
$$

**Assumption 13.** There exists a least favorable direction $g_0 \in \Theta$ such that for any $h \in \Theta$,

$$
\mathbb{E}\left\{\left(G_T\frac{p(H, \pi_e)}{p(H, \theta_0)} - s(H, \theta_0)[g_0]\right)s(H, \theta_0)[h]\right\} = 0.
$$

We make some remarks on these assumptions. Assumptions 9 and 10 impose restrictions on the sieve space, requiring the sieve space well approximate the parameter space. Such conditions hold for sieve space including B-spline and deep neural network. Assumptions 11 and 12 are commonly required in semi-parametric literature (Zhao & Zhang, 2017), restricting the complexity of function class around the true parameter. Assumption 13 indicates that there exists a projection of $\chi(H)p(H, \pi_e)/p(H, \theta_0)$ on the tangent space spanned by vector $s(H, \theta_0)[\cdot]$. This condition naturally holds when the parameter space is finite dimensional or the tangent space is a closed subspace.

**Proof of Theorem 9.**
We first show that for any $h \in \Theta$,

(i) $\sqrt{n}(\mathbb{P}_n - \mathbb{P})(s(H, \hat{\theta}_n)[h] - s(H, \theta_0)[h] = o_P(1)$.

(ii) $\mathbb{P}\left\{s(H, \theta_0)[h]\right\} = o_P(n^{-1/2}), \mathbb{P}_n\left\{s(H, \hat{\theta}_n)[h]\right\} = o_P(n^{-1/2})$.

For part (i), noted that $\mathbb{P}\left\{(s(H, \hat{\theta}_n)[h] - s(H, \theta_0)[h])^2\right\} = \mathbb{P}\left\{d^2(\theta_n, \theta_0)\right\} = o(1)$. Combining Assumption 11, the conclusion directly follows from Lemma 13.3 of Kosorok (2008). For part (ii), since $s(H, \theta)$ is the Fréchet derivative of log likelihood, it follows that $\mathbb{P}\left\{s(H, \theta_0)[h]\right\} = 0 = o_P(n^{-1/2})$. Meanwhile, Assumption 9 indicates that there exists $\tilde{h} \in \Theta_n$ such that $d(\tilde{h}, h) = o(n^{-1/4})$. Since $\theta_n$ maximize $\mathbb{P}_n L(H, \theta)$ in $\Theta_n$, it follows that $\mathbb{P}_n\left\{s(H, \hat{\theta}_n)[\tilde{h}]\right\} = 0$. Therefore, $\mathbb{P}_n\left\{s(H, \hat{\theta}_n)[h]\right\} = \mathbb{P}_n\left\{s(H, \hat{\theta}_n)[h] - s(H, \hat{\theta}_n)[\tilde{h}]\right\}$, which can be further decomposed into three parts;

$$
\begin{aligned}
\mathbb{P}_n\left\{s(H, \hat{\theta}_n)[h]\right\} &= (\mathbb{P}_n - \mathbb{P})\left(s(H, \hat{\theta}_n) - s(H, \theta_0)\right)[h] - (\mathbb{P}_n - \mathbb{P})\left(s(H, \hat{\theta}_n) - s(H, \theta_0)\right)[\tilde{h}] \\
&\quad + \mathbb{P}_n\left\{s(H, \theta_0)[h] - s(H, \theta_0)[\tilde{h}]\right\} \\
&\quad + \mathbb{P}\left\{(s(H, \hat{\theta}_n) - s(H, \theta_0))[h] - (s(H, \hat{\theta}_n) - s(H, \theta_0))[\tilde{h}]\right\} \\
&=: J_1 + J_2 + J_3.
\end{aligned}
$$

For $J_1$, follow a similar argument as proving claim (i), we obtain $J_1 = o_P(n^{-1/2})$. For $J_2$, $\mathbb{E}(\sqrt{n}J_2)^2 = O(d(h, \tilde{h})^2) = o(1)$, which indicates $J_2 = o_P(n^{-1/2})$. For $J_3$, direct calculation yields

$$
\mathbb{E}|J_3| \lesssim d(\theta_0, \hat{\theta}_n)d(h, \tilde{h}) = o(n^{-1/2}).
$$

Therefore, $\mathbb{P}_n\left\{s(H, \hat{\theta}_n)[h]\right\} = J_1 + J_2 + J_3 = o_P(n^{-1/2})$.

Combining claim (i),(ii) and Assumption 12, we obtain

$$
\begin{aligned}
\mathbb{P}_n\left\{s(H, \theta_0)[h]\right\} &= (\mathbb{P}_n - \mathbb{P})(s(H, \theta_0) - s(H, \hat{\theta}_n))[h] - \mathbb{P}_n\left\{s(H, \hat{\theta}_n)[h]\right\} \\
&\quad + \mathbb{P}\left\{s(H, \hat{\theta}_n)[h] - s(H, \theta_0)[h]\right\} \\
&= -\mathbb{P}\left\{\dot{s}_{\theta_0}[\hat{\theta}_n - \theta_0, h]\right\} + o_P(n^{-1/2}).
\end{aligned}
\tag{28}
$$

Take $h = \hat{\theta}_n - \theta_0$ in (28) yields

$$
\begin{aligned}
\mathbb{E}\left\{G_T \frac{p(H, \pi_e)}{p(H; \theta_0)}s(H, \theta_0)[\hat{\theta}_n - \theta_0]\right\} &= -\mathbb{E}\{s(H, \theta_0)[g_0]s(H, \theta_0)[\hat{\theta}_n - \theta_0]\} \\
&= \mathbb{E}\left\{\dot{s}_{\theta_0}[\hat{\theta}_n - \theta_0, g_0]\right\} \\
&= -\mathbb{P}_n\left\{s(H, \theta_0)[g_0]\right\} + o_P(n^{-1/2}).
\end{aligned}
\tag{29}
$$

Then, according to the Taylor expansion on $\theta_0$, we obtain

$$
\begin{aligned}
\widehat{v}_{\mathrm{IS}}(k) - \widehat{v}_{\mathrm{IS}}^{\dagger} &= \sum_{j=1}^{n} G_{i,T} \frac{p(H_j, \pi_e)}{p(H_j; \theta_0)} s(H_j, \theta_0)[\hat{\theta}_n - \theta_0] + O_P(\|\hat{\theta}_n - \theta_0\|^2) \\
&= \mathbb{E}\left\{ G_T \frac{p(H, \pi_e)}{p(H; \theta_0)} s(H, \theta_0)[\hat{\theta}_n - \theta_0] \right\} + o_P(n^{-1/2}) \\
&= -\mathbb{P}_n \left\{ s(H, \theta_0)[g_0] \right\} + o_P(n^{-1/2}),
\end{aligned}
$$

where the second equality holds because of Assumption 10.

Denote the main term on the right hand side be $\widehat{v}_3$. Then by Assumption 13, we have $\mathrm{Cov}(v_{\mathrm{IS}}^{\dagger}, \widehat{v}_3) = 0$. By the central limit theorem, $\mathrm{Var}(v_{\mathrm{IS}}^{\dagger})$ and $\mathbb{P}_n \left\{ s(H, \theta_0)[g_0] \right\}$ are of order $O(1/n)$. And thus, we have:

$$
\mathrm{Var}_A(\widehat{v}_{\mathrm{OIS}}(k)) = \mathrm{Var}_A(\widehat{v}_{\mathrm{OIS}}^{\dagger}) - \mathrm{Var}_A(v_3) \leq \mathrm{Var}_A(\widehat{v}_{\mathrm{OIS}}^{\dagger}),
$$

which completes the first inequality in Theorem 9.

Follow a very similar argument in proving $\mathrm{Var}_A(\widehat{v}_{\mathrm{OIS}}(k)) \leq \mathrm{Var}_A(\widehat{v}_{\mathrm{OIS}}^{\dagger})$, we can easily prove that $\mathrm{Var}_A(\widehat{v}_{\mathrm{SIS}}(k)) \leq \mathrm{Var}_A(\widehat{v}_{\mathrm{SIS}}^{\dagger})$ and $\mathrm{Var}_A(\widehat{v}_{\mathrm{DR}}(k)) \leq \mathrm{Var}_A(\widehat{v}_{\mathrm{DR}}^{\dagger})$, and hence, we omit the details of proof.

