# OpenReview forum: "Demystifying the Paradox of Importance Sampling with an Estimated History-Dependent Behavior Policy in Off-Policy Evaluation"
_ICML.cc/2025/Conference — ICML 2025 poster_

### Official Review · Reviewer_KUNq · 2025-03-14

**Overall Recommendation:** 3

**Summary:**

The paper provides a theoretical analysis of why estimating a history-dependent behavior policy in off-policy evaluation (OPE) can reduce mean squared error (MSE). The authors derive a bias-variance decomposition for OPE estimators and show that history-dependent behavior policy estimation reduces variance at the cost of increasing finite-sample bias. Theoretical results establish that the variance reduction is monotonic as history length increases, except for the Marginalized Importance Sampling (MIS) estimator, which worsens with more history. Empirical results on CartPole validate these theoretical findings.

## update after rebuttal
The authors have addressed my concerns in the rebuttal. I appreciate the additional clarifications and intuitions provided. I no longer have major concerns with the submission and have updated my score accordingly.

**Claims And Evidence:**

1. The paper claims that history-dependent behavior policy estimation leads to variance reduction in OPE estimators, which is well-supported by both theoretical derivations and empirical results.
2. The claim that this variance reduction comes at the cost of increased finite-sample bias is also well-grounded, as the paper provides a bias-variance decomposition to justify this trade-off.
3. However, the paper does not formally establish when the increased bias outweighs variance reduction, making it unclear how to choose an optimal history length in practical applications.

**Essential References Not Discussed:**

Liu and Zhang (2024) published at ICML studies offline-informed behavior policy selection and is directly related to this paper’s topic. This work is a representative of the various  non-parametric behavior policy estimation methods.

**Experimental Designs Or Analyses:**

1. The empirical results confirm the theoretical findings, showing that variance decreases and bias increases as history length grows.
2. The lack of details on experimental setup makes reproducibility difficult—there is no appendix detailing implementation choices, hyperparameters, or sampling strategies.
3. There are no novel insights provided in the experiment section.

**Methods And Evaluation Criteria:**

1. The paper does not introduce a new method or algorithm but instead provides an analytical perspective on existing estimators.
2. The choice of CartPole as an evaluation environment is somewhat limited, as prior OPE work typically includes MuJoCo tasks to assess generalization.

**Other Comments Or Suggestions:**

1. The literature review should include non-parametric behavior policy estimation in a more structured way.
2. A more detailed discussion on practical implications—such as guidance on choosing history length—would make the results more practical.

**Other Strengths And Weaknesses:**

Strengths:
1.The projection-based interpretation offers a useful mathematical lens on the variance reduction effect.

Weaknesses:
1. The paper does not introduce a new method, and the analysis itself is not particularly novel, as the variance reduction effect is well understood in the context of importance sampling.

**Questions For Authors:**

See above.

**Relation To Broader Scientific Literature:**

1. The paper builds on prior work in off-policy evaluation (OPE) and importance sampling, particularly extending prior bias-variance analysis in OPE settings.
2. There is a line of research on non-parametric behavior policy estimation that has already been demonstrated to outperform parametric methods in various environments, yet this is not acknowledged.

**Theoretical Claims:**

1. The variance reduction property of history-dependent estimation is well-supported by the derived bias-variance decomposition.
2. The paper introduces a projection-based interpretation, which is conceptually interesting but follows naturally from standard variance-reduction techniques.
3. While the theoretical results explain when variance is reduced, they do not explicitly analyze how estimation errors in the learned behavior policy affect bia.

---

> ### Author Rebuttal · Authors · 2025-03-29
>
> **Choice of History Length**  Excellent comment. We fully agree that optimal selection of history length is crucial for applying our theory to practice. In response, we have **developed a method during the rebuttal, supported by promising simulation results**. Our approach is motivated by the bias-variance trade-off revealed in our theory: while increasing history length reduces asymptotic variance for OIS/SIS/DR estimators, it might increase finite-sample bias. We therefore propose to select the history length that minimizes $h^* = \arg\min_h [2n\widehat{\text{Var}}(h) - h\log(n)]$ where:
> - $\widehat{\text{Var}}(h)$ denotes variance estimator computed via the sampling variance formula or bootstrap;
> - $k\log(n)$ is the BIC penalty (Schwarz, 1978) preventing selecting long history without substantial reduction of the variance.
>
> [Results](https://www.dropbox.com/scl/fi/02eppzq8qpjygt4cxmc28/SelectKBIC.png?rlkey=k30z87thebot3la7apnvsob11&st=wvta6lho&dl=0) show that in all cases, OIS estimators with our adaptively selected history achieve the lowest MSE compared to those using fixed history.
>
> **No novel insights from experiments**.  The primary aim of our paper is to provide a rigorous theoretical analysis of how history-dependent behavior policy estimation affects the bias and variance of OPE estimators. Our main contribution lies in establishing these theories, through the derived bias-variance trade-offs. Accordingly, our experiments are to empirically verify these theories rather than to derive new insights. Indeed, our experimental results align with the theory.
>
> **MuJoCo**. As suggested, we conducted simulations in MuJoCo Inverted Pendulum environment during rebuttal. [Results](https://www.dropbox.com/scl/fi/egiz1bjz3pztlqs5p1lzb/mujoco.jpg?rlkey=qrg7wjltut5tcpbhlurspligd&st=vmssm274&dl=0) again, align with our theory.
>
> **No new method**. First, as noted in the ICML 2025 Call for Papers, "Theory of Machine Learning" is a core research area -- in parallel to RL, deep learning, and optimization. Our paper falls within this category.
>
> Second, while not introducing new method, our theoretical analysis offers useful guidelines to practitioners. Table 1 shows that history-dependent behavior policy estimation should be used with OIS, SIS, DR estimators with misspecified Q-function, but may be unnecessary for DR with correct Q-functions or MIS estimators.
>
> Third, we did develop a new method for history length selection during rebuttal and obtained promising empirical results (see our response #1). We are happy to use the extra page to present this method shall our paper be accepted.
>
> **Variance reduction effect in IS**. We respectfully clarify a potential misunderstanding regarding our theoretical contributions. While the benefits of **designing** optimal proposal distributions for IS (pre data collection) are indeed well-established (Liu & Zhang, 2014) and connected to the literature on optimal experimental design for policy learning (Agarwal et al., 2019) and policy evaluation (Hanna et al., 2017; Mukherjee, 2022; Li et al., 2023), our work addresses a different problem: the theoretical benefits of **estimating** such distributions (post data collection). In other words, we did not consider policy design, but study how history-dependent behavior policy estimation impacts OPE — a question only empirically explored (and solely for OIS estimators) in prior work (Hanna, 2019, 2021). To our knowledge, our analysis provides the first theoretical foundation for these empirical observations.
>
> **The lack of experimental details**. We would like to make some clarifications:
>
> * We detailed the DGP and the episode length in Appendix A.1. All data were generated by this  DGP **without additional sampling strategies**.
> * As for implementation, we mentioned the use of logistic regression for behavior policy estimation in Appendix A.1. To be more specific, we employed **scikit-learn’s LogisticRegression with all hyperparameters kept at their default values** (no custom tuning).
> * During rebuttal, we have created an [anonymous repository](https://www.dropbox.com/scl/fi/8hxbqti3t9yu1boeb7u7e/code.zip?rlkey=yq1adjgw84cmc20va59rsc1o1&st=5022v5pc&dl=0) containing all the code for implementation.
>
> **Essential reference**. While we are happy to include the paper by Liu and Zhang (2024), we respectfully disagree that it is an essential reference. This paper is about the **design** of behavior policies whereas we study **estimating** such policies. While the reviewer describes their behavior policy as "non-parametrically estimated," we find no discussion of nonparametric estimation in this paper.
>
> Regarding nonparametric methods more broadly: while potentially relevant and worthwhile to cite, they are not central to our focus on history-dependent estimation. We included Kallus & Uehara (2020) who employed history-dependent behavior policy estimation to handle history-dependent target policies. Happy to include other references

---

> > ### Comment · Reviewer_KUNq · 2025-04-03
> >
> > Thank you for your detailed and thoughtful rebuttal, as well as the additional experiments and clarifications. I appreciate the effort you put into addressing the concerns raised.
> >
> > That said, I believe some issues remain. In particular, I believe the omission of recent work such as Liu & Zhang (ICML 2024) is an oversight. While I understand your distinction between policy estimation and design, both works fundamentally tackle the question of learning behavior policies from offline data to improve OPE. A discussion of this connection would have helped better situate your contribution within the broader literature.
> >
> > Additionally, while the theoretical analysis is rigorous and clearly presented, the key insight regarding the bias-variance trade-off is relatively intuitive and builds upon prior empirical observations. The additional method for selecting history length and the extended experiments provided during the rebuttal are helpful and strengthen the work.
> >
> > Overall, I believe the paper would benefit from a clearer positioning within related literature and a more thorough empirical evaluation to complement the theoretical analysis. I hope the authors will consider these suggestions to further improve the paper.

---

> > > ### Author Response · Authors · 2025-04-08
> > >
> > > Thank you for recognizing the value of our additional experiments and our newly proposed methodology for history selection, as well as for acknowledging the difference between policy design and estimation. We greatly appreciate the opportunity to respond again and address your remaining comments.
> > >
> > > **Liu & Zhang (ICML 2024)**. As mentioned in our response, we are happy to include the paper. Specifically, we plan to include the following discussion in the Discussion Section:
> > >
> > > > We note that a separate line of research (Hanna et al., 2017; Mukherjee, 2022; Li et al., 2023; Liu & Zhang, 2024) investigates optimal experimental design for off-policy evaluation (OPE). These works focus on **designing** optimal behavior policies prior to data collection to improve OPE accuracy whereas our proposal considers **estimating** behavior policies after data collection for the same purpose. The work of Liu & Zhang is particularly related as the behavior policy is computed from offline data before being run to collect more data. Both approaches share the most fundamental goal of enhancing OPE by learning behavior policies — whether for data collection or retrospective estimation.
> > >
> > > We hope this addresses your comment.
> > >
> > > **Theoretical insights**. We respectfully argue that our work provides novel theoretical insights beyond what has been empirically observed in the existing literature. While prior empirical studies (Hanna, 2019, 2021) demonstrated variance reduction through history-dependent behavior policy estimation for OIS, we systematically study three other estimators in addition to OIS, corresponding to SIS, DR, MIS.
> > >
> > > More importantly, our analysis reveals that history-dependent behavior policy estimation yields fundamentally different effects across the three estimators:
> > >
> > > 1. For SIS, it reduces the variance;
> > > 2. For DR, variance reduction occurs when the Q-function is misspecified, while the variance remains unchanged under correct Q-function specification;
> > > 3. For MIS, it inflates the variance.
> > >
> > > These findings have not been systematically documented in prior empirical studies, nor have they been theoretically analyzed in existing literature.
> > >
> > > **Empirical evaluation**. We greatly appreciate this comment. Although our paper is primarily theoretical, we have conducted extensive empirical studies in response to your comment. These newly obtained results are organized into three parts:
> > >
> > > 1. Investigation of the performance of adaptive history selection (refer to our response during rebuttal);
> > > 2. Evaluation of OIS across three MuJoCo environments (results reported in this [figure](https://www.dropbox.com/scl/fi/myea6xplhzi6irv1r0hto/OIS.pdf?rlkey=pnwmg3zb459f9f4n4ned60za2&st=o9v82yjk&dl=0)):
> > >    - (i) Inverted Pendulum (with continuous action space);
> > >    - (ii) Inverted Double Pendulum (with a higher state dimension than (i));
> > >    - (iii) Swimmer (a substantially different environment from both (i) and (ii));
> > > 3. Evaluation of SIS, DR, and MIS in the Swimmer environment (results reported in this [figure](https://www.dropbox.com/scl/fi/wfvsqhgktok69i6tojlmq/swimmer.pdf?rlkey=irmok3tppls6fm8nx23bableu&st=6l4j7n2v&dl=0)).
> > >
> > > We are happy to include these experiments, as well as any additional experiments the reviewer may suggest, in the final version of the paper should it be accepted, to directly address your comment.

---

### Official Review · Reviewer_Hj4i · 2025-03-14

**Overall Recommendation:** 3

**Summary:**

The paper discusses a paradox in offline policy evaluation through importance sampling, where the performance of the target policy is estimated from a weighted average of the reward value by the ratio of the target policy and the behavioral policy. The paper suggests that the mean-squared error of the said estimator can be improved if the behavioral policy is estimated in a broader family of models. For example, if the true behavior probability is known, the paper suggests that people should replace it with the empirically estimated behavior probability. If the true behavior policy is context-independent, people should estimate it as if it were context-dependent. Finally, if the true behavior policy is Markovian, people should estimate it as if it were a higher-order Markovian function. The authors made an analogy to doubly robust estimator, though I could not fully understand the details.

**Claims And Evidence:**

I am not convinced with the proofs. I cannot follow the proof of Lemma 1, let alone the rest of the paper.

**Essential References Not Discussed:**

N/A

**Experimental Designs Or Analyses:**

No.

**Methods And Evaluation Criteria:**

The experiments are not clearly presented. The authors gave a numerical example in Section 3.1 with additional details in Appendix A, but they left out key details regarding the numerical values of the mean-squared errors of the estimator.

**Other Comments Or Suggestions:**

N/A

**Other Strengths And Weaknesses:**

N/A

**Questions For Authors:**

I cannot follow Lemma 1 as the conclusions appear counter-intuitive. I would appreciate it if the authors could elaborate on the proof of Lemma 1. Can the authors prove it from first principles without using advanced methods like Neyman orthogonality?

**Relation To Broader Scientific Literature:**

I did see papers suggesting that approximating the behavior policy offers empirical advantages over using the true behavior policy. However, I have always speculated that it has to do with clipping effects, where the estimated behavioral policies are regularized to prevent extreme values. The authors seem to have other intuitions that I am not familiar with.

**Theoretical Claims:**

I cannot follow the proof of Lemma 1 in Appendix B.

---

> ### Author Rebuttal · Authors · 2025-03-29
>
> We thank the reviewer for their feedback regarding the proof of Lemma 1. We acknowledge that its reliance on Neyman orthogonality may not be immediately familiar to general audience. Apart from this concept, the proof employs standard techniques, including basic calculus such as Taylor expansions. We highlight that our original proof is clear and mathematically sound. The reviewer’s concerns arise from the misunderstanding of these technical tools rather from any technical flaw in the proof itself. Having said that, we have provided an alternative proof without Neyman orthogonality below.
>
> **Proof of Lemma 1**. Define $n(a)$ as the number of times action $a$ is taken, $n(s,a)$ as the number $(s,a)$ pairs in the dataset, and $n(s)=\sum_a n(s,a)$. We first prove the second inequality $\textrm{MSE}_A(\widehat{v}\_{\text{IS}}^{\text{CA}})\le \textrm{MSE}_A(\widehat{v}\_{\text{IS}}^{\dagger})$. For any estimator $\widehat{v}$, it follows from the law of total variance that $$\text{Var}(\widehat{v})  = \underbrace{\mathbb{E}(\text{Var}(\widehat{v}| \\{n(a)\\}\_{a} ))}\_{I} + \underbrace{\text{Var}(\mathbb{E}(\widehat{v}|\\{n(a)\\}\_{a} ))}\_{II}.$$ In the following, we will show that:
>
> (i) **Term I**: The difference between $\widehat{v}\_{\text{IS}}^{\dagger}$ and $\widehat{v}\_{\text{IS}}^{\text{CA}}$ is negligible;
>
> (ii) **Term II**: $\widehat{v}\_{\text{IS}}^{\dagger}$ achieves a larger value than $\widehat{v}\_{\text{IS}}^{\text{CA}}$.
>
> Combining (i) and (ii) yields that $\text{Var}\_A(\widehat{v}\_{\text{IS}}^{\dagger})\ge  \text{Var}\_A(\widehat{v}\_{\text{IS}}^{\text{CA}})$. As both estimators are asymptotically unbiased, we obtain $\text{MSE}\_A(\widehat{v}\_{\text{IS}}^{\dagger})\ge \text{MSE}\_A(\widehat{v}\_{\text{IS}}^{\text{CA}})$.
>
> Specifically, by direct calculation, $$\mathbb{E}\\{\text{Var}(\widehat{v}\_{\text{IS}}^{\dagger}|\\{n(a)\\}\_a )\\}=\mathbb{E}\Big(\frac{1}{n}\sum_{a\in\mathcal{A}} \frac{\pi_e^2(a)}{\pi_b^2(a)}\frac{n(a)}{n}\sigma_a^2\Big),\quad \mathbb{E}\\{\text{Var}(\widehat{v}\_{\text{IS}}^{\text{CA}}|\\{n(a)\\}\_{a} )\\} = \mathbb{E}\Big(\frac{1}{n}\sum_{a\in\mathcal{A}} \frac{\pi_e^2(a)}{(n(a)/n)^2}\frac{n(a)}{n}\sigma_a^2\Big).$$
> According to the law of large numbers, $n(a)/n$ is to converge in probability to $\pi_b(a)$. It follows that
>          $\mathbb{E}\{\text{Var}(\widehat{v}\_{\text{IS}}^{\dagger}|\\{n(a)\\}\_{a} )\} - \mathbb{E}\{\text{Var}(\widehat{v}_{\text{IS}}^{\text{CA}}|\\{n(a)\\}\_{a} )\} = o(1/n)$. This proves (i). On the other hand,
>
> $$\mathbb{E}(\widehat{v}\_{\text{IS}}^{\text{CA}}|\\{n(a)\\}\_{a}) = \frac{1}{n}\mathbb{E} \Big(\sum_{a\in\mathcal{A}} \frac{\pi_e(a)}{n(a)/n}\cdot n(a)\mathbb{E}[R|A=a] \Big) = \mathbb{E}\Big(\sum_{a\in\mathcal{A}} \pi_e(a)\mathbb{E}[R|A=a] \Big)$$
> is independent of $n(a)$. Consequently, we have $\text{Var}\\{\mathbb{E}(\widehat{v}\_{\text{IS}}^{\text{CA}}|\\{n(a)\\}\_{a})\\}=0$.
>  However,
> $$\text{Var}\Big(\mathbb{E} \\{\widehat{v}\_{\text{IS}}^{\dagger}|\\{n(a)\\}\_{a} \\}\Big) = \text{Var}\left(\frac{1}{n}\sum_{a\in\mathcal{A}}\frac{\pi_e(a)}{\pi_b(a)}n(a)\mathbb{E}[R|A=a]\right)$$
> is dependent of $n(a)$.  This verifies (ii). Meanwhile, notice that $\text{MSE}\_A(\widehat{v}\_{\text{IS}}^{\dagger})\ge \text{MSE}\_A(\widehat{v}\_{\text{IS}}^{\text{CA}})$ if and only if $ \text{Var}\\{\mathbb{E}(\widehat{v}\_{\text{IS}}^{\dagger}|\\{n(a)\\}_{a}) \\}=0$, which indicates that $\mathbb{E}(R|A)=0$ almost surely.
>
> The first inequality $\textrm{MSE}\_A(\widehat{v}\_{\text{IS}}^{\text{CD}})\le \textrm{MSE}\_A(\widehat{v}\_{\text{IS}}^{\text{CA}})$ can be similarly proven. Due to space constraints, we present only a proof sketch. Applying the law of total variance again, we obtain
> $$\text{Var}(\widehat{v})  = \mathbb{E}(\text{Var}(\widehat{v}| \\{n(s,a)\\}\_{s,a} )) + \mathbb{E}(\text{Var}(\mathbb{E}(\widehat{v}| \\{n(s,a)\\}\_{s,a} ) | \\{n(a)\\}\_{a} )) + \text{Var}(\mathbb{E}(\widehat{v}|\\{n(a)\\}\_{a} )).$$
> Similarly, we can show that
>
> (i) The difference in the first term between the two estimators is asymptotically negligible.
>
> (ii) $\widehat{v}\_{\text{IS}}^{\text{CD}}$ achieves a smaller second term (zero), as its conditional expectation is independent of $\\{n(s,a)\\}\_{s,a}$.
>
> (iii) The conditional expectations of both estimators are independent of $\\{n(a)\\}\_{a}$, so the last term is zero for both estimators.
>
> Consequently, $\widehat{v}\_{\text{IS}}^{\text{CD}}$ achieves a smaller asymptotic variance, and equivalently, MSE.
>
> **Clipping effects**. Our theory is not about clipping making the estimation of the behavior policy desirable for OPE. We dedicated an entire section (Section 3) to build intuition. As can be seen in the first two equations on Page 4, estimating the behavior policy effectively transforms the original IS estimator into a doubly robust estimator, which is known to outperform standard IS estimators when the Q-/reward function is well-approximated. This explains the benefits of such an estimation.

---

> > ### Comment · Reviewer_Hj4i · 2025-04-05
> >
> > The authors addressed my concerns. I appreciate the intuitions in the proofs and have no major concerns. On a minor side, how is the proof connected with Neyman orthogonality?

---

> > > ### Author Response · Authors · 2025-04-08
> > >
> > > We are delighted to hear that our responses have addressed your comments and sincerely appreciate your increase in our score.
> > >
> > > The proof we provided during the rebuttal is a version without Neyman orthogonality, while our original proof involves Neyman orthogonality. To elaborate how it connects to Neyman orthogonality, we remind you that there are three key steps in our original proof:
> > >
> > > - The first step is to show that the IS estimators $\widehat{v}\_{\text{IS}}^{\text{CA}}$ and $\widehat{v}_{\text{IS}}^{\text{CD}}$ with estimated IS ratios are equivalent to the DR estimators with estimated IS ratios and reward functions; see the last two equations on Page 12. This follows directly from basic calculus, leveraging the fact that both $\widehat{r}$ and $\widehat{\pi}_b$ are derived using tabular methods.
> > >
> > > - The second step is to show that these DR estimators, although involving **estimated** IS ratios and reward functions, are asymptotically equivalent to DR estimators with **oracle** IS ratios and reward functions. This is where Neyman orthogonality comes into play.
> > >
> > > - The last step is to directly compare the MSEs of these DR estimators with oracle IS ratios and reward functions against $\widehat{v}_{\text{IS}}^{\dagger}$ to demonstrate the advantages of (context-dependent) behavior policy estimation. This step, again, follows directly from basic calculus.
> > >
> > > We provided a detailed discussion of Neyman orthogonality, including its definition, usage, and the mathematical details of the second step, in our response to Referee qCNN (see both our rebuttal and post-rebuttal responses). In summary, when applied to our setting, Neyman orthogonality can be understood as a property of OPE estimators. An OPE estimator achieves this property if its expected value is robust to small perturbations in the estimation errors of the reward and behavior policy near their oracle values. Specifically, these estimation errors affect the OPE estimator’s mean only in second order. This ensures that the OPE estimator’s bias decays much faster than the estimation errors in the reward and behavior policy, making it asymptotically negligible.
> > >
> > > According to the proofs of Theorems 5.1 & 5.2 in Chernozhukov et al. (2018), the DR estimator satisfies Neyman orthogonality. As a result, it is safe to replace the estimated IS ratios and reward functions in DR with their oracle values without introducing significant bias. Meanwhile, the asymptotic variance of DR remains unchanged, as long as the estimated reward and behavior policy are consistent. Consequently, its MSE also remains asymptotically unchanged.

---

### Official Review · Reviewer_qCNN · 2025-03-19

**Overall Recommendation:** 4

**Summary:**

The paper investigates the bias-variance trade-off and the MSE in IS-based off-policy evaluation. A first part of the paper investigates bandits and shows that context-conditioning and visitation-count based approximation of the behavior policy can reduce the MSE (even though the policy is context-independent). They then evaluate different estimators for RL policies with their estimated non-Markovian behavior policy, and show this leads to reduced variance compared to non-Markovian behavior policy (longer history length leading to lower variance but exponentially growing bias) and improved asymptotic MSE. They show this principle works across different base-estimators, including OIS, Sequential IS, Doubly Robust and MIS estimators.

# update after rebuttal

The authors have clarified some concepts and done additional experiments. Therefore I keep my score of accept (4).

**Claims And Evidence:**

The authors provide proof for all their claims, although some more background can be discussed for the reader to appreciate these.

**Essential References Not Discussed:**

In general, there are not that many references from recent years, e.g. in the sequential IS and doubly robust techniques. The paper also does not expand much on the relation to these techniques. While the technique is a plugin for a variety of estimators, it does make sense to discuss the relation to these other works. In particular, a more in depth discussion of the related techniques would clarify the contribution.
Conditional IS [1] conditions on random variables in the trajectory. Techniques which modify the importance ratio are even more directly related to your work, since modifying the behavior policy changes the importance ratio. In this category, it makes sense to discuss State-based IS [2], which is similar in spirit that it is a plugin that can be applied to (the same) variety of estimator classes (OIS, Sequential IS, MIS, DR). It modifies the ratio based on whether the states in the history contribute to the return or not. Similarly, the already included PDIS and INCRIS references can be discussed in more depth, as I would argue in some cases these are equivalent to your technique (see questions for authors).

[1]  Rowland, M., Harutyunyan, A., Hasselt, H., Borsa, D., Schaul, T., Munos, R. & Dabney, W.. (2020). Conditional Importance Sampling for Off-Policy Learning. Proceedings of the Twenty Third International Conference on Artificial Intelligence and Statistics (AISTATS 2020) 108:45-55.

[2] Bossens, D.M., & Thomas, P.S. (2024). Low Variance Off-policy Evaluation with State-based Importance Sampling. IEEE Conference on Artificial Intelligence (CAI 2024), 871-883.

**Experimental Designs Or Analyses:**

The analysis uses a cartpole domain. to verify the bias and the MSE. The results are in line with the theoretical claims.

The implementations of the algorithms in the theory and especially the experiments could be more clearly described with additional details and more specific references. For instance, the implementation of the base-algorithms, any specific hyperparameters, and the details of the non-Markovian behavior policy estimation.

**Methods And Evaluation Criteria:**

The authors design a technique which is applicable to a wide variety of estimators and clearly demonstrate its benefit in reducing the MSE in the asymptotic case while also discussing limitations, e.g. the finite sample bias.

**Other Comments Or Suggestions:**

“The re-weighted returns are then averaged to produce an unbiased estimator of the target policy’s value.” This is not true for all IS based estimators. Perhaps you can stress it is ordinary importance sampling.

$\pi_e / \pi_b \leq C$ --> forget to mention for all $(s,a)$ ?

 I presume in Eq. 4, it should be $\hat{v}_{\text{SIS}}$.

in discussion, “can increases” --> can increase

**Other Strengths And Weaknesses:**

Strengths:

- The problem setting is interesting

- The paper is rigorous and well-presented

- The objectives are clear

- The theoretical and empirical results align

Weaknesses:

- In some cases, a bit more background of the techniques may need to be provided for a general audience.

- The related works discussion could be expanded on, and maybe some more recent works could be included.

**Questions For Authors:**

Appendix B.: “According to Neyman orthogonality, both the estimated reward and estimated behavior policy can be asymptotically replaced  by its oracle value (Chernozhukov et al., 2018) without changing the OPE estimator’s asymptotic MSE.” Can the authors provide explain this step? Is the Neyman orthogonality always valid?

What is the relation between history dependent behavior policies and history-dependent importance ratios (sequential importance sampling)? I believe that your technique can be considered as a generalisation (or special case, depending how we look at it), of sequential importance sampling techniques. OIS has history length 1, incremental importance-sampling has history length k (for chosen k), and per-decision importance sampling has all past time steps until current time t as history. In your framework, the importance ratio is also history dependent. It will be interesting to see what the intuition is for the finding that a lower number of time steps of the ratio is beneficial for reducing the variance while a higher number of time steps in the behavior policy is beneficial for reducing the variance, and what is actually the key difference if any, between these two formulations. Is it fair to say that when applying a non-Markovian behavior policy to OIS, it is equivalent to per-decision importance sampling in case using all past time steps of the episode, and equivalent to incremental importance sampling when using past $k$ times steps of the episode? In short, a non-Markovian behavior policy and the importance ratio definition in sequential IS techniques seems highly related, and possibly interchangeable in the OIS case, but this relation is currently not exactly clear.

"Alternatively, the k-step history $H_{t−k:t}$ can be used
to construct a history-dependent MIS ratio $wt (k) = E(λt |H_{t−k:t} , A_t )$ ..To appreciate why Theorem 8 holds, notice that by setting
k to the horizon $T$ , $w_t(k)$ is reduced to the $λ_t$ , and the
resulting estimator is reduced to SIS, which suffers from
the curse of horizon and is known to be less efficient than
MIS." Following this and my above remark, perhaps a similar analysis can be done to derive the relation between the OIS/SIS variants with non-Markovian $\pi_b$ and the sequential IS algorithms.

**Relation To Broader Scientific Literature:**

The work seems to be closely related, but complementary to, other techniques for variance reduction in OPE, such as per-decision IS, weighted IS, incremental IS, conditional IS, and state-based IS. The difference to these techniques is that here the estimator's variance reduction is based on using an estimated non-Markovian behavior policy. It seems related to Sequential IS methods (e.g. PDIS and INCRIS), depending on the time window used, when combined with OIS as the backbone, but this relation is not expanded on in the text. However, the authors do show that their algorithm applied to MIS can make it comparable to SIS for $k=T$.

**Theoretical Claims:**

The theoretical claims are backed up by extensive, well-presented proofs. I unfortunately, could not check them all. The theorems are additionally confirmed in the experiments.

---

> ### Author Rebuttal · Authors · 2025-03-30
>
> Many thanks for your excellent comments and positive assessment of our paper. We will include these references, use the extra page to expand the related work and address all minor comments. In the following, we focus on clarifying your questions.
>
> **Neyman Orthogonality**. This property enables us to establish the asymptotic equivalence between an OPE estimator that uses estimated reward and behavior policy, and the one that employs their oracle values. Specifically, the difference between the two estimators can be decomposed into the following three terms:
>
> $$\mathbb{E}_n\Big( \sum_a \pi_e(a|S)[\widehat{r}(S,a)-r(S,a)]- \frac{\pi_e(A|S)}{\pi_b(A|S)}[\widehat{r}(S,A)-r(S,A)] \Big)+\mathbb{E}\_n\Big[ \Big(\frac{\pi\_e(S,A)}{\widehat{\pi}_b(S,A)} - \frac{\pi_e(S,A)}{\pi_b(S,A)}\Big)[R- r(S,A)] \Big]+\mathbb{E}_n \Big(\frac{\pi\_e(S,A)}{\widehat{\pi}_b(S,A)} - \frac{\pi_e(S,A)}{\pi_b(S,A)}\Big)[\widehat{r}(S,A)-r(S,A)] ,$$
>
> where $\widehat{r}$ and $\widehat{\pi}_b$ denote the estimated reward and behavior policy, respectively. For the moment, let us assume these estimators are computed based on some external dataset. Then:
>
> * The first two terms are of zero mean. They are of the order $o_p(n^{-1/2})$ provided that $\widehat{r}$ and $\widehat{\pi}_b$ converge to their oracle values.
> * The last term is of the order $\|\|\widehat{r}-r\|\| \times \|\|\widehat{b}-b\|\|$ where $\|\|\widehat{r}-r\|\|$ and $\|\|\widehat{b}-b\|\|$ denote the root MSEs (RMSEs) between $\widehat{r}(S,A)$ and $r(S,A)$, and between $\widehat{b}(S,A)$ and $b(S,A)$, respectively. Crucially, the order is the product of the two RMSEs. Consequently, as they decay to zero at a rate of $o_p(n^{-1/4})$ -- which is much slower than the parametric rate $O_p(n^{-1/2})$ -- this term becomes $o_p(n^{-1/2})$ as well.
>
> Consequently, the difference is $o_p(n^{-1/2})$, which establishes the asymptotic equivalence between the two estimators.
>
> When $\widehat{r}$ and $\widehat{\pi}_b$ are estimated from the same dataset, the orders of the three terms additionally depend on the VC indices measuring the complexity of the reward and behavior policy models. Returning to our bandit example, we employ tabular methods to estimate both nuisance functions. Given the finite state and action spaces, their VC dimensions are finite as well. Additionally, both RMSEs achieve the parametric convergence rate $O_p(n^{-1/2})$. This validates our claim.
>
> **History-dependent behavior policy & history-dependent IS ratios**. Our opinion is that (i) reducing the time horizon in the IS ratio and (ii) increasing the history length in the estimation of the behavior policy are two generally different approaches to improving OPE performance, although these methods are related when considering MIS (which we will discuss in detail later). Specifically:
>
> * Approach (i) is more aggressive (in certain cases), aiming to reduce the **order** of the variance and address the curse of horizon by using shorter-history IS ratios. For example, the incremental IS (IIS) estimator you mentioned uses a fixed history length $k$. A small $k$ can reduce the estimator's variance from exponential in $T$ to polynomial ($O(T^k)$). Conceptually, this shifts the original per-decision IS (PDIS) estimator toward the MIS estimator, with IIS representing an intermediate point between PDIS and MIS. However, this comes at the cost of increased bias due to the ignored long-term dependencies.
> * Approach (ii) is more conservative, targeting the **magnitude** (rather than order) of the variance by incorporating longer history in behavior policy estimation. As our theory shows, the variance preserves the same order (and thus the curse of horizon remains). According to our bandit example, this approach effectively converts a standard IS into a DR. In contrast to approach (i), it introduces minimal bias, as proven in our theory.
>
> We also remark that these approaches can be combined to doubly reduce IS estimator's variance. First, one may specify a history length k for the IS ratio to obtain the IIS estimator. Next, the IIS ratio can be estimated using history-dependent behavior policy estimation to further reduce the variance magnitude.
>
> However, the two approaches do interact when it comes to MIS, which incorporates the IS ratio of both the state and action, rather than actions alone. Consequently, using history-dependent behavior policy estimates ratios over complete histories rather than individual states alone, which becomes equivalent to employing history-dependent IS ratios.
>
> During the rebuttal, we have created a [figure](https://www.dropbox.com/scl/fi/pl5j53c60z7im0inbkj9j/ISRelation.jpg?rlkey=gydxh3bpo3hew8h7zuhg4kzox&st=htnlbaix&dl=0) to visualize their interactions. Specifically, applying history-dependent behavior policy estimation to MIS can yield IIS (when ignoring state ratios beyond k steps), OIS, or PDIS. To the contrary, reducing the history-dependence in IS ratios converts OIS to PDIS and PDIS to IIS

---

> > ### Comment · Reviewer_qCNN · 2025-04-03
> >
> > Actually, reducing the time horizon does not change the order of the estimator's variance to polynomial. It just reduces the exponent from $T$ to $k < T$. In [3], there is no analysis of INCRIS that shows a change to polynomial order. There is however, an analysis of the options framework (Corollary 1 and 2), which is not the same but also reduces the number of timesteps, but which reduces the exponent rather than making it polynomial. In Eq.8 of [2], reducing the number of time steps reduces the exponent too, not making it polynomial. I also add  the reference for PDIS [4]. What is the authors' statement based on?
> >
> > So both the order and the technique seem to be closely related. I do think it is possible, as the authors mention, that both types of approaches can be applied together.
> >
> > I find the authors' explanation for Neyman orthogonality and the proof incomprehensible:
> > - The property's effect is explained but the property itself is not explained.
> > - The factor b is not introduced.
> > - Multiplying $n^{-1/2}*n^{-1/2}$ does not give me $n^{-1/4}$ but $n^{-1}$
> >
> > [3] Guo, Thomas, & Brunskill (2017). Using Options and Covariance Testing for Long Horizon Off-Policy Policy Evaluation. NeurIPS 2017.
> >
> > [4] Precup,Sutton, & Singh (2000), “Eligibility Traces for Off-
> > Policy Policy Evaluation,” ICML 2000.
> >
> > Edit: Additionally, I was wondering the following. It is not so surprising to obtain similar results with the Inverted Pendulum vs Cartpole. As shown in https://gymnasium.farama.org/environments/mujoco/inverted_pendulum/, they are the same except that the action space is now continuous. They have a very specific reward structure not shared by other environments, Is there any reason why two very similar problems are chosen?

---

> > > ### Author Response · Authors · 2025-04-08
> > >
> > > We apologize for any lack of clarity, typos and the space constraint in our initial response that may have caused confusion. We sincerely appreciate the opportunity to respond again and provide further clarification.
> > >
> > > **Neyman orthogonality (in response to your first bullet point)**. Let us start by clarifying Neyman orthogonality, introduced by Chernozhukov et al. (2018, The Econometrics Journal). This property is named after the famous statistician Jerzy Neyman, who used such properties in constructing efficient estimators and hypothesis tests (Neyman, 1959, Probability and Statistics: The Harald Cramér Volume). Since its introduction, it has been widely employed for robust and efficient estimation and inference in econometrics, statistics and machine learning, with applications ranging from the estimation of heterogeneous treatment effect (Oprescu et al., 2019, ICML), off-policy evaluation in RL (Kallus and Uehara, 2022, Operations Research), and variable selection (Quinzan et al., 2023, ICML).
> > >
> > > The original definition involves two parameters: a primary parameter of interest $\theta$ and a nuisance parameter $\eta$. Suppose we have an estimating equation $\phi(O,\eta,\theta)$ with $O$ being the data observation, such that $\theta$ can be estimated by solving $\mathbb{E}_n[\phi(O,\eta,\theta)] = 0$. The Neyman orthogonality ensures that $\phi$ is robust to small perturbations in $\eta$ near the true value $\eta_0$. Mathematically, it requires:
> > >
> > > $$
> > > \nabla_{\eta} \mathbb{E} \phi(W; \theta_0, \eta) \Big|_{\eta = \eta_0} = 0,
> > > $$
> > >
> > > where $\nabla_{\eta}$ represents the Gateaux derivative evaluated at the true parameters $\eta_0$ and $\theta_0$. This implies that estimation errors in $\eta$ affect the expectation of $\phi$ only through second-order terms.
> > >
> > > Back to our bandit example, $\theta$ corresponds to the target policy's value $v(\pi_e)$, $\eta$ corresponds to the reward function $r$ and the behavior policy $\pi_b$ that need to be estimated, $O$ represents the context-action-reward triplet $(S, A, R)$, and $\phi$ reduces to the difference between the estimating function $\psi(O; \eta)$ and $v(\pi_e)$. For instance, when we use DR for estimation,
> > >
> > > $$
> > > \psi_{\text{DR}}(O; r, \pi_b) = \sum_a \pi_e(a|S) r(a, S) + [R - r(A, S)] \frac{\pi_e(A|S)}{\pi_b(A|S)}.
> > > $$
> > >
> > > By definition, requiring $\phi$ to satisfy Neyman orthogonality is equivalent to requiring $\psi$ to satisfy such condition. This condition indeed holds for the doubly robust estimating equation $\psi_{\text{DR}}$; see, e.g., Step 1 of the proofs of Theorems 5.1 & 5.2 in Chernozhukov et al. (2018, The Econometrics Journal). This is precisely why in our response, the impact of estimation error on the last term can be expressed as a product $\\|\widehat{r} - r\\| \times \\|\widehat{\pi}_b - \pi_b\\|$, as the estimation error in $r$ and $\pi_b$ affects the policy value only at second order.
> > >
> > > **Rate clarification (in response to your third bullet point)**. To establish the asymptotic equivalence, the DR with estimated reward and behavior policy must be close to the estimator using oracle reward and behavior policy, up to an error of $o_p(n^{-1/2})$. Due to the second-order effect, we typically require both $\widehat{r}$ and $\widehat{b}$ to converge at a rate of $o_p(n^{-1/4})$.
> > >
> > > In the tabular settings, these estimators converge at a faster rate of $O_p(n^{-1/2})$. Consequently, as you mentioned, the resulting error is $O_p(n^{-1})$, which is sufficiently small to meet our $o_p(n^{-1/2})$ requirement. In contrast, without Neyman orthogonality, the estimator would be sensitive to first-order errors in the nuisance estimators. In that case, the final error would be $O_p(n^{-1/2})$ (big-$O_p$), which does not meet the required $o_p(n^{-1/2})$ (little-$o_p$) condition.
> > >
> > > **Notation clarification (in response to your second bullet point)**. Apologize for the typo. The symbol $b$ shall be replaced with the behavior policy $\pi_b$.
> > >
> > > **Variance reduction**. We completely agree that the variance of PDIS remains exponential in $k$. In our response, we referred to the use of a substantially small $k$ — specifically, $k$ chosen proportional to $K \log(T)$ — so that the overall error is reduced to polynomial order in $T$, i.e., $O(T^K)$. We apologize for not making this point clearer.
> > >
> > > **Empirical Evaluation**. In the rebuttal, we used the Inverted Pendulum environment as Reviewer KUNq requested experiments within the MuJoCo framework. While we acknowledge that this environment is similar to CartPole, it serves to examine settings with continuous actions. Though our primary contribution is theoretical analysis, in this round, we have expanded our empirical analysis to include a wider range of more complex MuJoCo environments (e.g., Swimmer). Refer to our response to Reviewer KUNq for details.

---

### Decision · Program_Chairs · 2025-05-01

**Decision:**

Accept (poster)

**Comment:**

The paper presents a theoretical analysis of history-dependent behavior policy estimation in off-policy evaluation (OPE), demonstrating that it reduces variance at the cost of increased finite-sample bias. The authors provide both theoretical derivations and empirical validation on CartPole. The rebuttal addressed reviewers' concerns, offering additional clarifications and intuitions that have helped resolve initial doubts. However, the paper still lacks practical guidance on selecting the optimal history length, and the experimental evaluation remains limited to CartPole, with insufficient details on implementation choices, making reproducibility difficult.

The theoretical contributions are sound but not entirely novel, as variance reduction via history-dependent behavior policies is well-established. Nonetheless, the paper is a valuable addition to the literature, offering clear insights into bias-variance trade-offs in OPE. While the paper could benefit from more detailed practical implications and a broader experimental evaluation, the rebuttal has alleviated major concerns. Therefore, I recommend acceptance.